# Mechanisms of sensorimotor adaptation in a hierarchical state feedback control model of speech

**Kwang S. Kim** [1] *, **Jessica L. Gaines** [2], **Benjamin Parrell** [3],
**Vikram Ramanarayanan** [4,5], **Srikantan S. Nagarajan** [6], **John F. Houde** [4]

**1** Department of Speech, Language, and Hearing Sciences, Purdue University, West Lafayette, Indiana, United States of America, **2** Graduate Program in Bioengineering, University of California Berkeley-University of California San Francisco, San Francisco, California, United States of America, **3** Department of Communication Sciences and Disorders, University of Wisconsin–Madison, Madison, Wisconsin, United States of America, **4** Department of Otolaryngology-Head and Neck Surgery, University of California San Francisco, San Francisco, California, United States of America, **5** Modality.AI, San Francisco, California, United States of America, **6** Department of Radiology and Biomedical Imaging, University of California San Francisco, San Francisco, California, United States of America

* kwangkim@purdue.edu

**Data Availability Statement:** The source code and simulation configurations presented in the study can be accessed on a public GitHub repository (https://github.com/kwangsk/FACTS). All

## Abstract

Upon perceiving sensory errors during movements, the human sensorimotor system updates future movements to compensate for the errors, a phenomenon called sensorimotor adaptation. One component of this adaptation is thought to be driven by sensory prediction errors–discrepancies between predicted and actual sensory feedback. However, the mechanisms by which prediction errors drive adaptation remain unclear. Here, auditory prediction error-based mechanisms involved in speech auditory-motor adaptation were examined via the feedback aware control of tasks in speech (FACTS) model. Consistent with theoretical perspectives in both non-speech and speech motor control, the hierarchical architecture of FACTS relies on both the higher-level task (vocal tract constrictions) as well as lower-level articulatory state representations. Importantly, FACTS also computes sensory prediction errors as a part of its state feedback control mechanism, a well-established framework in the field of motor control. We explored potential adaptation mechanisms and found that adaptive behavior was present only when prediction errors updated the articulatory-to-task state transformation. In contrast, designs in which prediction errors updated forward sensory prediction models alone did not generate adaptation. Thus, FACTS demonstrated that 1) prediction errors can drive adaptation through task-level updates, and 2) adaptation is likely driven by updates to task-level control rather than (only) to forward predictive models. Additionally, simulating adaptation with FACTS generated a number of important hypotheses regarding previously reported phenomena such as identifying the source(s) of incomplete adaptation and driving factor(s) for changes in the second formant frequency during adaptation to the first formant perturbation. The proposed model design paves the way for a hierarchical state feedback control framework to be examined in the context of sensorimotor adaptation in both speech and non-speech effector systems.

simulation data presented in the current study can be found on an OSF data repository (DOI: 10. 17605/OSF.IO/W37HV).

**Funding:** This research was funded by grants from the National Institute of Health (F32DC019538 to K. S.K., R01DC017696, R01DC017091, R01DC013979, R01NS100440, and P50DC019900 to J.F.H., S.S.N.) and an Achievement Rewards for College Scientists (ARCS) Foundation award to J.L. G. The funders had no role in study design, data collection and analysis, decision to publish, or preparation of the manuscript.

**Competing interests:** The authors have declared that no competing interests exist.

## Author summary

When we move, our brain predicts the sensory feedback that would result from the movement, and can quickly adjust future movements based on any sensory prediction errors— differences between the predictions and actual sensory feedback. This learning process, sensorimotor adaptation, has been extensively studied in many movements (e.g., walking, reaching, speaking), but its underlying mechanisms remain largely unclear. Here, we examined mechanisms driving speech adaptation in response to altered auditory feedback using the FACTS model, a hierarchical state feedback control model of speech in which a high-level controller achieves speech goals (e.g., constrictions of the vocal tract) by directing a low-level controller that moves the speech articulators (e.g., positions of the jaw and the tongue). We demonstrated that prediction errors can drive adaptation through changes in high-level control, but not solely through changes in predictions of movement outcomes or low-level control. In addition to replicating multiple key features of sensorimotor adaptation in speech, our simulations also generated potential new explanations for phenomena that are currently poorly understood. Importantly, given that our model design is closely aligned with widely accepted motor control frameworks outside of speech, these results have the potential to be broadly applicable to non-speech motor systems as well.

## Introduction

The human sensorimotor system has the remarkable ability to update future movements in response to environmental changes. In the context of speech production, such learning abilities have been well demonstrated by *auditory-motor* adaptation, in which participants modify their subsequent speech production after their auditory feedback is experimentally altered by shifting vowel formants (vocal tract resonances that distinguish vowels from one another) or vocal pitch. Speech auditory-motor adaptation remains a topic of great interest (see [1] for an extensive review on formant adaptation studies) and has been utilized to investigate speech motor control in both healthy (e.g., [2–4]) and clinical populations (e.g., cerebellar ataxia, [5]; Parkinson's disease, [6]; stuttering, [7]).

Although sensorimotor adaptation in speech has been extensively investigated, the underlying mechanisms driving this learning are not firmly established. According to studies of upper limb reaching movements, visuo-motor adaptation (i.e., adaptation to perturbed visual feedback) can be driven by two types of errors: sensory prediction error and performance (or task) error. Sensory prediction error is the discrepancy between the actual sensory consequences and those predicted based on motor efference copy (e.g., [8]). It is thought that sensory prediction errors contribute to the *implicit* component of learning, which occurs without learner awareness. On the other hand, performance error (or task error) is the discrepancy between the outcome of the movement and the intended goal or target of the movement. This type of error occurs with the awareness of the learner and involves strategy use, contributing to the *explicit* component of learning (e.g., [9]). More recent studies have, however, demonstrated that task errors can also drive implicit learning ([10–15], also see Discussion).

Interestingly, whereas both components of learning contribute to visuo-motor adaptation, there are multiple lines of evidence suggesting that speech auditory-motor adaptation may be driven entirely by the implicit component. The most convincing evidence for the implicit

nature of auditory-motor adaptation is that participants are completely unaware of their adaptive behaviors (e.g., [16–18]). In fact, participants show adaptation even when they are specifically instructed to avoid changing their productions [17] and continue to be unaware of their adaptation even when asked, after every utterance, whether and how they are changing their speech [19]. Lametti and colleagues [20] also discovered that even when cognitive load was increased by a dual-task paradigm, presumably affecting ongoing explicit processes, the amount of auditory-motor adaptation remained unchanged even as visuo-motor adaptation was affected. The implicit nature of auditory-motor adaptation is also supported by the findings demonstrating that speech adaptation can be removed by adding even a small amount of delay (e.g., 100 ms) in auditory feedback [21, 22], given that adding delays in sensory feedback has been shown to disturb implicit learning, but not explicit learning [23–25]. Based on these pieces of evidence, multiple research groups have postulated that auditory prediction errors—discrepancies between predicted and actual auditory feedback—drive auditory-motor adaptation (e.g., [19, 20, 26, 27]).

However, prediction errors themselves cannot yield adaptation if the controller does not change its future productions. Thus, a crucial mechanism involved in sensorimotor adaptation is how error signals influence the controller (i.e., the mechanism that is responsible for generating appropriate motor output) to issue modified or learned motor outputs in the subsequent movements. One possibility is that sensory errors drive corrective motor commands which are then integrated into the controller for future productions [28–31]. However, sensory errors can update future behaviors without online feedback-based correction in both speech [7] and non-speech [32, 33] adaptation.

Previous studies have also suggested that the controller may be an inverse of an internal model that can predict sensory consequences based on a given motor command (i.e., an inverse of a forward model) and it may be updated to produce learned behaviors whenever its paired forward model is updated from error signals (e.g., [34]). An alternative perspective is that sensorimotor adaptation can be driven by updating the controller directly with some type of learning (or error) signals. According to this idea, the controller's control policy (or control law) may be directly updated from the learning signals. Even though this learning mechanism was originally thought to be responsible for other types of motor learning such as model-free learning (see [35]), a recent upper limb adaptation study using mirrored feedback suggested that direct updates to the control policy, but not updates to the forward models, can explain adaptation behavior [36]. In the context of speech auditory-motor adaptation, it remains largely unclear whether the implicit adaptation process involves forward model-based updates or direct control policy updates. To date, only one study has examined whether the paired inverse and forward models could yield auditory-motor adaptation via a computational model [37], but due to its simplified proof-of-concept design, the model could not simulate a detailed account of sensorimotor processes involving prediction errors and state estimation.

Although it remains possible that other types of errors such as task errors may also drive implicit speech adaptation, as an initial step toward understanding mechanisms underlying the learning process, we investigated potential mechanisms through which auditory prediction errors can drive adaptation using our recently introduced computational model of speech, FACTS (Feedback-Aware Control of Tasks in Speech, [38]). FACTS is a fully developed model of real-time sensorimotor control in speech based on state feedback control [39, 40], a well-established framework in the field of general (non-speech) motor control (e.g., [41–45], but also see [46] for speech production). Crucially, the predictive nature of the state feedback control architecture in FACTS allows us to directly examine the idea that adaptation is driven by sensory prediction errors.

## FACTS model overview

FACTS was developed as a hierarchical model that relies on both the higher-level task representations (e.g., vocal tract constrictions) as well as lower-level "body-state" or articulatory representations (e.g., articulator positions) [38]. Such a hierarchical framework is well-grounded in the speech motor control literature which suggests that higher-level tasks defined in terms of the location and size (or "degree") of constrictions in the vocal tract effectively control speech articulators, minimizing the degrees of the freedom (e.g., [47, 48]). Congruent with the literature, all FACTS designs tested in the current study employed both the task and articulatory state (see Materials and methods for more details).

A total of three different design variants of FACTS were implemented and tested to simulate auditory-motor adaptation. In all designs, internal models were updated when auditory prediction errors were detected; the designs differed with respect to which internal model modules were updated by these errors. The first two model designs (Fig 1) had a similar architecture to our previously proposed model [38]. In the first design (Design A), auditory prediction errors updated the articulatory state estimate ($\tilde{\mathbf{a}}_t$) through Kalman gain-based state correction ($\mathcal{K}\Delta y_t$). In turn, the corrected articulatory state estimate was used to update the

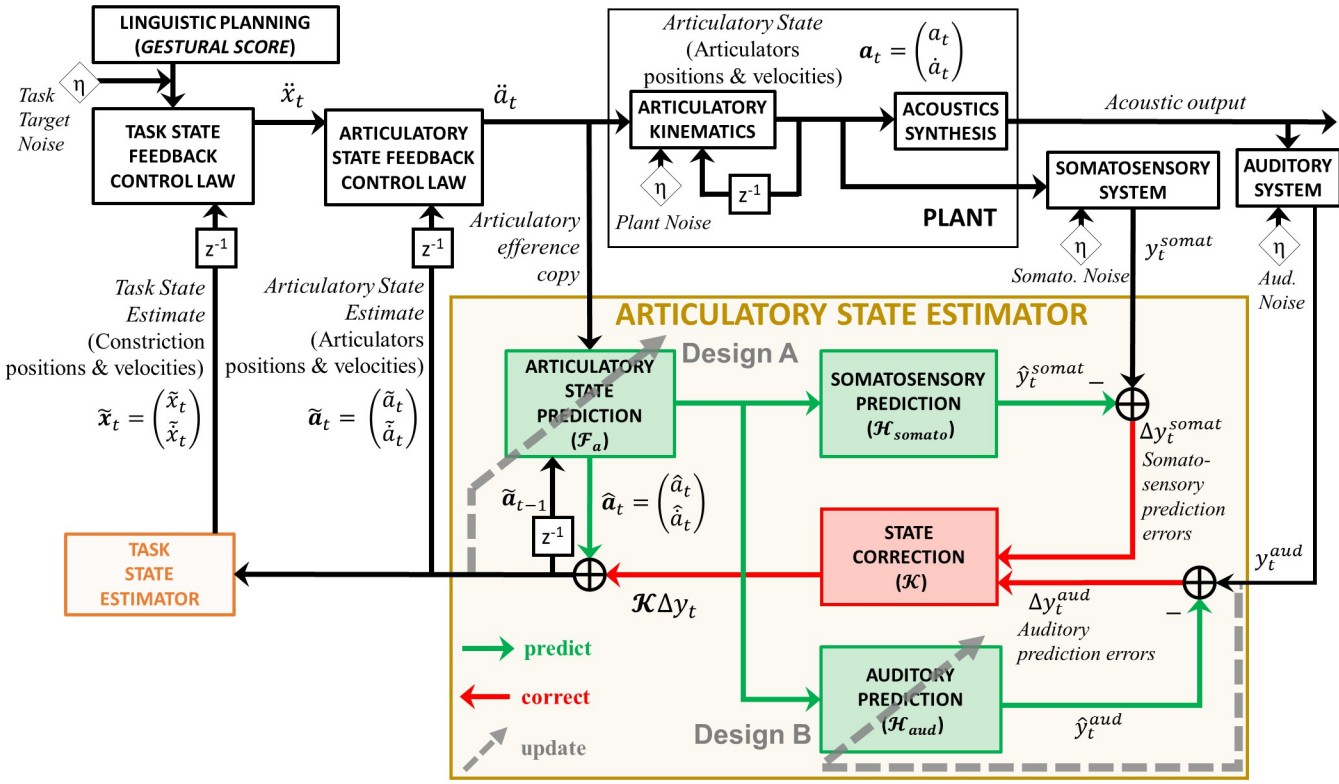

**Fig 1. Designs A and B were developed based on the original FACTS architecture in Parrell et al. [38].** Design A updates the articulatory state prediction based on the final state estimate ($\tilde{a}_t$) which is determined by the original state prediction ($\hat{a}_t$) and state correction signals based on the auditory prediction errors ($\kappa_t \Delta y_t$). Design B updates the auditory forward model (i.e., auditory prediction) directly from the auditory feedback. Note that there are a few differences between these models and the original design [38], which are described in the Materials and methods section in detail. $\eta$ denotes Gaussian noise. For adaptation simulations (which involved a short utterance), we did not allow any online compensation, simulating the fact that auditory feedback-based online compensation begins 100–200 ms after the onset of the perturbation (e.g., [27, 50–52]) and somatosensory feedback-based compensation would be nearly negligible (since no somatosensory perturbation was applied). Thus, the default state feedback pathways that allow online compensation were disabled. Instead, the prediction-based pathways were used.

forward model that predicted the current articulatory state. The articulatory forward model, which generated predictions for each articulatory dimension, was trained and updated using Locally Weighted Projection Regression (LWPR, [49]). The second design (Design B) used the same architecture as Design A, but implemented adaptation by updating only the auditory forward model, rather than the state prediction. In this design, LWPR was used to update the auditory forward model using the auditory feedback.

The last design (Design C) used a modified hierarchical architecture, inspired by a recently proposed idea that there may be segregated loops that characterize cortical-subcortical connections for motor control at the task and "body" levels [53]. In this design, therefore, we implemented separate state feedback control loops for the articulatory level and the task level. This modified hierarchical architecture was also motivated by the fact that auditory feedback is most informative about higher-level vocal tract tasks (e.g., a high first formant is associated with a narrow constriction at palate), whereas it is less relevant to the lower-level articulatory state (i.e., positions of all speech articulators/muscles, see [54]). Hence, Design C expanded on the task state estimation process previously implemented as a simple transformation from the articulatory state estimate, such that a second "observer" estimated the task state based on the previous articulatory state, an efference copy of the task-state motor command, and auditory feedback (Fig 2). Specifically, the articulatory state estimate of the previous time step was transformed to a task state representation via an articulatory-to-task transformation model learned using Locally Weighted Projection Regression (LWPR). The transformed previous task state, along with an efference copy of the task state motor command, was sent to the task state prediction module to generate a task state prediction. An Unscented Kalman Filter (UKF, [55]) process then computed the final estimate of task state ($\tilde{\mathbf{x}}_t$) based on the task state prediction ($\hat{\mathbf{x}}_t$) and auditory feedback. To model adaptation, the final (corrected) task state estimate was used to update the articulatory-to-task transformation LWPR model (Fig 2).

In all three designs, we attempted to simulate adaptation in response to perturbations in the first formant frequency (F1) with repeated trials of a short production (150 ms duration) of /ɛ/. For adaptation data, the first and second formant frequencies (F1 and F2) were extracted from the middle portion of the vowel (from 50 ms to 100 ms after the vowel onset). Given that auditory feedback-based within-utterance compensation is known to begin ∼100–200 ms after perturbation onset (e.g., [27, 50–52]), such online compensation response can be considered to be minimal in this context. Additionally, even though the latency of a somatosensory feedback-based corrective response is known to be shorter than that of auditory (e.g., [56, 57]), the application of an auditory perturbation alone would not involve somatosensory prediction errors, so somatosensory feedback-based corrective response would also be negligible. Therefore, in all adaptation simulations presented in this study, both auditory and somatosensory feedback pathways for *online* compensation were disabled. Rather, the sensory feedback or its related information (i.e., state correction signals) were stored and used to update relevant modules *after* each movement, if the size of the prediction errors exceeded a given threshold (see below).

We modeled adaptation as trial-to-trial updates to internal predictive Locally Weighted Projection Regression (LWPR) models (Design A and B) and a transformation LWPR model (Design C). Within each trial, whenever auditory prediction errors were detected, the prediction error-based state estimate was stored in an error memory buffer. After each trial, this memory buffer was used to update predictive/transformation LWPR models, and the updated LWPR models were used on the next trial. Another feature that was implemented for all designs was auditory prediction error detection threshold. The auditory prediction error in F1 was measured against this threshold to determine whether such error

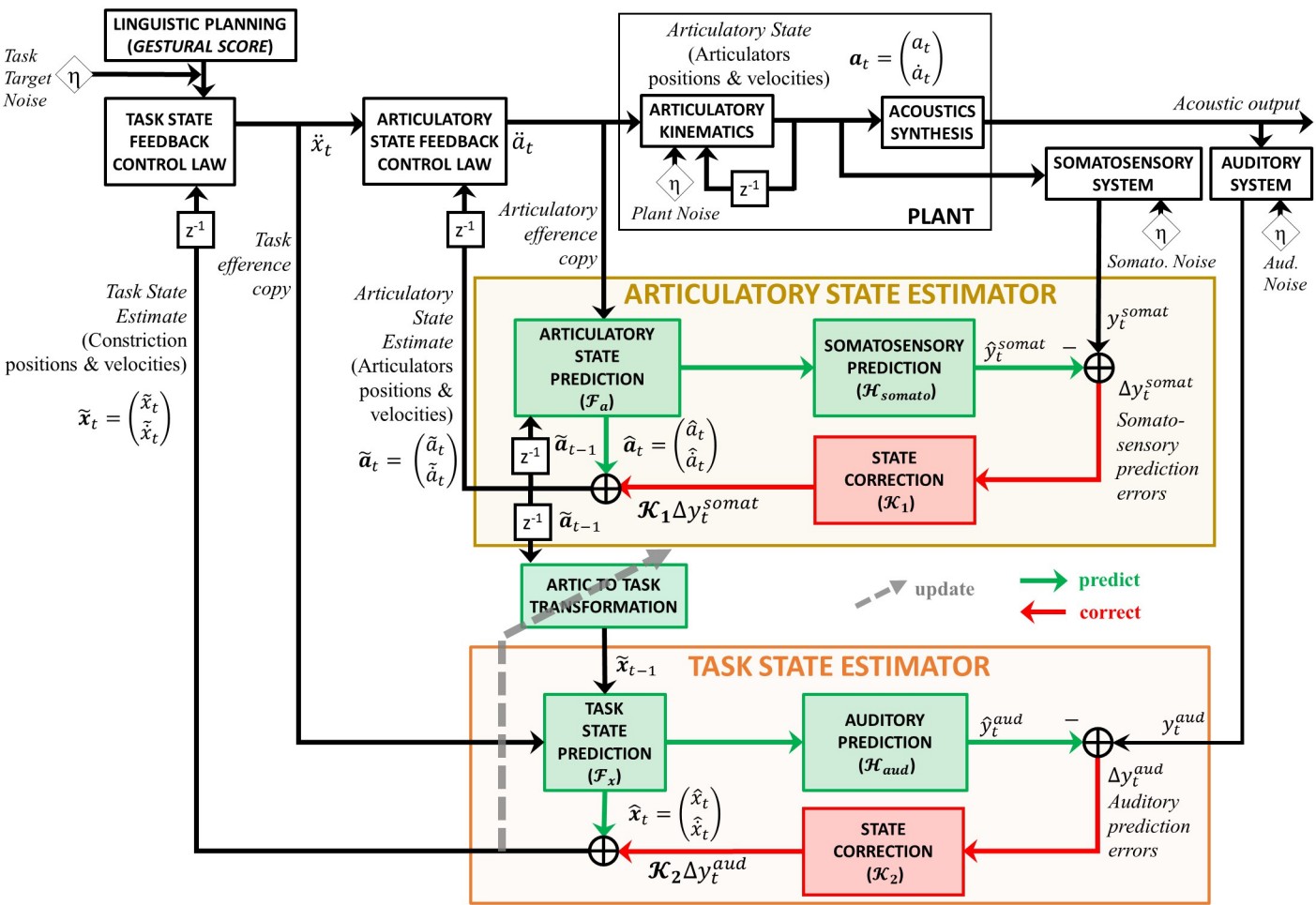

**Fig 2. Design C was developed based on a modified hierarchical FACTS architecture.** For Design C, a modified hierarchical FACTS architecture in which auditory feedback is used by the task state estimator, rather than the articulatory state estimator, was used. The task state estimator generates a task state prediction based on the previous articulatory state estimate via an articulatory-to-task transformation LWPR model and an efference copy of the task motor commands. This task state prediction can be corrected using auditory prediction errors. During adaptation simulations, when auditory prediction errors are detected, the final (corrected) estimate is used to update the articulatory-to-task transformation model. $\eta$ denotes Gaussian noise. As with Design A and B, sensory feedback-based online compensation was disabled.

should be "remembered." If the error was larger than the given threshold, the error was registered for various module updates shown in Figs 1 and 2, providing opportunities for yielding adaptation. More in-depth descriptions are included in the Materials and methods section.

In addition to attempting to simulate adaptation that matches with empirical data, mechanisms driving the adaptive behavior were examined by altering different modules of FACTS during simulations (e.g., fixing auditory prediction). Additionally, given that the modified hierarchical architecture is novel, its online compensation in response to within-utterance unpredictable formant perturbation was also investigated to ensure that the new architecture was able to replicate the previously reported compensatory behavior (for the original architecture's online compensation simulations, see [38]). Finally, we also explored how changes to various model parameters (e.g., auditory noise, auditory prediction error detection threshold, and task target noise) affected adaptation.

## Results

### Updating forward articulatory state or sensory prediction models does not cause adaptation

In Design A, articulatory state correction signals that arose from auditory prediction errors updated the articulatory state prediction model, which is a forward model that predicts the current articulatory state from the previous state and an efference copy of the articulatory-level motor commands. Although this design slightly lowered the first formant frequency (F1) in response to the 400 cents up-shift in F1 (Fig 3, top left), the extent of this decrease was almost negligible ($\sim$5 Hz over 140 trials). Moreover, this change resulted from a slow, steady growth

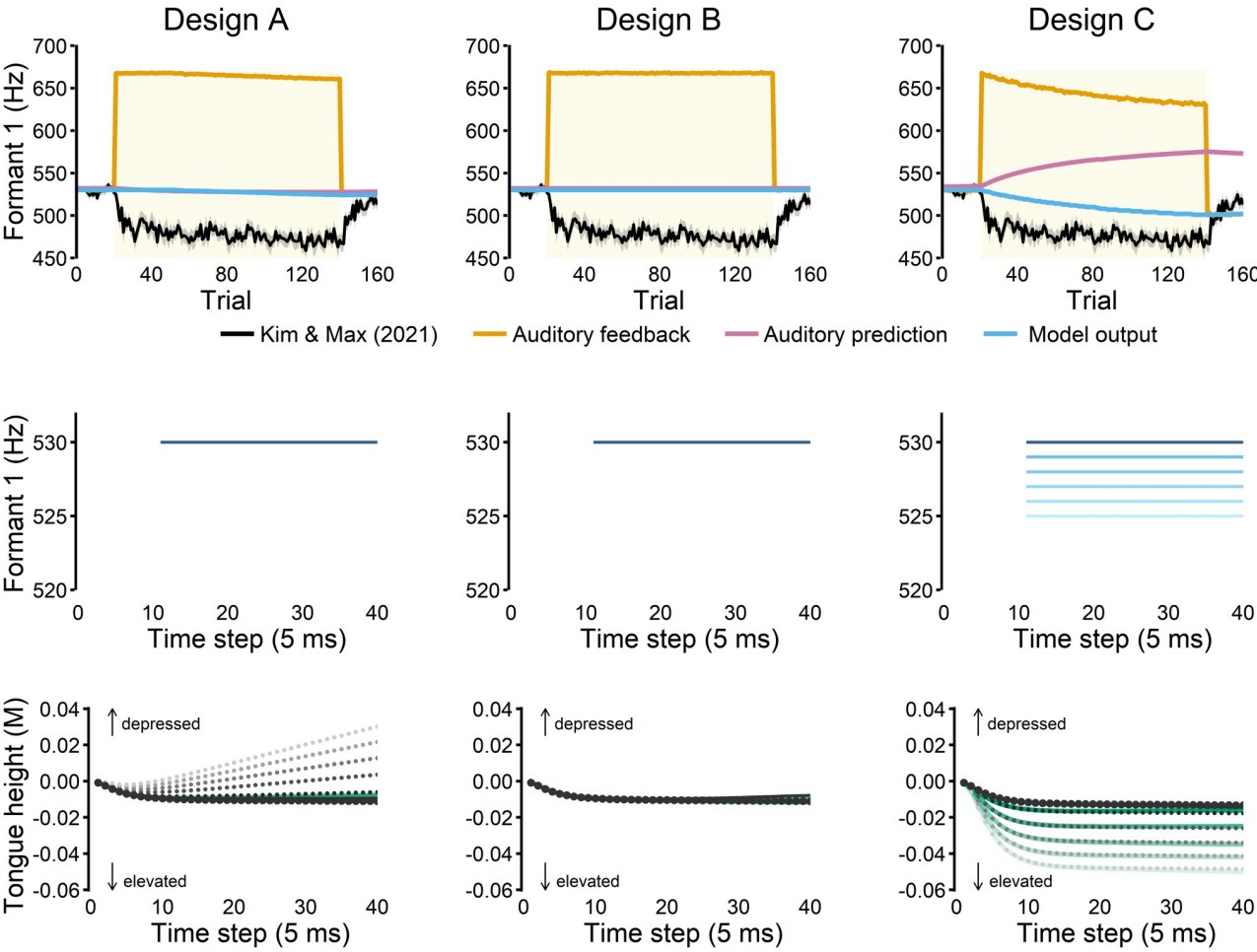

**Fig 3. Simulation results for different model designs. Top row**: A 400 cents up-shift in the first formant frequency (F1) was applied from trial 21 to trial 140 (yellow shaded area). To extract F1 for each trial, we averaged F1 of the middle 10 time steps (time step 11 to 20) of the 30 time steps of the simulated acoustic data for each production. Experimental data was retrieved from the control group in Kim & Max [19]. **Middle row**: F1 values produced across time during five early perturbation trials (trials 22, 24, 26, 28, and 30). Lighter shades indicate later trials. Note that the first 10 time steps for each trial are pre-phonatory preparatory movements from the model's default start position, so no acoustic data are plotted. **Bottom row**: The true articulatory state of tongue height (solid green lines) and its state estimate (gray dots), expressed in the Maeda Principal Component unit (M), plotted across time steps for the early perturbation trials (trials 22, 24, 26, 28, and 30). Black dots indicate the estimate in a baseline trial. Lighter shades indicate later trials. In Design A, the estimate diverged from the true state across the time steps, and the amount of divergence also gradually increased across the trials. Only in Design C, the true articulatory state for the tongue height demonstrated noticeable adaptation across trials (green lines). The estimates (gray dots) closely tracked their true state in Design C.

over trials rather than commonly reported rapid adaptation response (e.g., [19]). In fact, no change in F1 was visible in the first 10 perturbation trials (see Fig 3, middle left). The articulatory state data across time steps showed that the articulatory state estimates started to diverge from the true state as the forward articulatory prediction model was updated (Fig 3, bottom left), likely driven by the up-shift F1 perturbation causing the estimator to underestimate the true tongue height. Nevertheless, the true state (i.e., the actual movement) changed minimally across the perturbation trials, leading to minimal changes in the acoustics.

We also explored several other variations of Design A, but none of them simulated adaptation. Relying solely on auditory feedback (by removing somatosensory feedback), for example, resulted in a behavior that first looked like adaptation, but each vowel production had unstable behavior in which the articulators never reached a stable final position. Such within-vowel formant changes are not consistent with empirical data. In addition, applying another type of Kalman filter also did not solve this problem. These results demonstrated that Design A, regardless of other factors such as the amount of reliance on auditory feedback, was not able to simulate adaptation (see S1 Appendix for more details).

In Design B, the forward auditory prediction model was updated based on actual (perturbed) auditory feedback whenever auditory prediction errors were detected. Although the simulation showed that auditory prediction was indeed updated due to the perturbation (i.e., predicting higher F1, see Design B in Fig 3), neither the model's acoustic nor articulatory output showed any changes from the baseline behavior. These results indicated that updates in auditory prediction alone did not generate adaptation in FACTS. Although it has been shown that updates to forward models in other frameworks can yield adaptation (e.g., [58]), it is important to note that such models used updated forward models to influence the motor command, leading to adaptation. In Design B, forward model updates did not directly affect motor commands, and our results clearly demonstrated that such a model design did not yield adaptation.

## Updating the task state transformation model produces realistic adaptation

The poor performance of Designs A and B indicated that a different architecture might be necessary to yield adaptive behavior. First, the simulations from Design A suggested that changes at the task level, as opposed to the articulatory level, might be necessary to produce adaptation. Second, the simulation results from Design B demonstrated that updating the forward auditory prediction model alone did not yield any noticeable changes in the control laws, suggesting that an update in task state, which can affect the task state feedback control law, might be necessary. However, the architecture for Designs A and B did not include any methods to directly update task state.

Hence, inspired by recent proposals of hierarchical feedback control for human movement [53], we developed an architecture in which auditory feedback was integrated in the task state, rather than the articulatory state estimation system (Design C). In this modified hierarchical architecture, auditory prediction errors directly influenced the task state by updating the learned articulatory-to-task state transformation. We hypothesized that updating this transformation may yield adaptive changes, given the critical role of the task state estimate in the feedback control law.

In Design C, the final state estimate, which was corrected based on auditory prediction errors, was used to update the articulatory-to-task transformation model after each utterance. The simulation results showed a gradual decrease in F1 across trials, although with a slower rate of learning and smaller extent compared to the experimental data (Fig 3, top and middle

right). Additionally, the articulatory estimate was very close to the true articulatory state across the perturbation trials in the simulation (Fig 3, bottom right). Together, these findings indicated that updating the articulatory-to-task transformation model produced adaptive behavior broadly consistent with auditory-motor adaptation.

Nonetheless, the simulated adaptation was smaller and slower than adaptation observed in the experimental data. The small adaptation response of the model was likely due to the Kalman filter which assumed a high magnitude of random noise in the sensory signals in order to achieve stable control. Thus, when faced with auditory perturbation, the filter attributed most of the prediction error resulting from the perturbation to random noise in the auditory feedback and relied primarily on the original state prediction to estimate the final estimated state. Therefore, the final estimate determined by the Kalman filter was closer to the original prediction, yielding only a small adaptation response. Although it was possible to maximize the contribution of sensory input to the final state estimate by adjusting Kalman filter variables (e.g., reducing the magnitude of the sensory covariance matrix or increasing the magnitude of the prior covariance matrix), such changes were less than optimal as they introduced substantial instability in the final estimate. That is, because the estimator was tuned to react to any changes in the measurement, the estimator also unnecessarily reacted to regular noisy sensory signals, even without any consistent perturbation. Such behaviors were undesirable as the Kalman filter then became unstable and sub-optimal in an unperturbed environment.

Numerous control applications involving sudden and large changes in the environment or the controller have achieved stable control using *adaptive* Kalman filters (e.g., controlling an unmanned aerial vehicle under sensor or actuator failures in [59]). In a common adaptive Kalman filter design, residual errors (prediction errors inversely weighted by their measurement noise covariance) are constantly monitored. If they become larger than a certain threshold, the Kalman filter can adapt so that the estimator becomes more sensitive to the sensor information (e.g., auditory feedback) and the resulting Kalman gain becomes larger. This large Kalman gain allows greater and faster updates in the state estimate, outperforming non-adaptive Kalman filters. Importantly, if the residual error is smaller than this threshold (e.g., in an unperturbed environment), the Kalman filter does not increase its sensitivity to the measurements, which offers stable tracking. Adaptive Kalman filters, therefore, are more versatile across different environments compared to non-adaptive Kalman filters.

In order to simulate more realistic sensorimotor adaptation, we implemented a simple adaptive unscented Kalman filter (adaptive UKF or AUKF) in Design C (see Fig 4A). At each time step, the task estimator computed $\epsilon$, the square of the auditory prediction errors, inversely weighted by the auditory noise covariance matrix. If $\epsilon$ was larger than a given threshold ($\gamma$), the AUKF parameters were increased by multiplying gains. Specifically, the cross covariance matrix (cross covariance of the task state and auditory feedback sensory measurements sigma points), $\mathbf{P_{xy}}(t)$ was increased 6-fold from the default. In addition, the process noise matrix ($\mathbf{Q}(t-1)$) and the prior covariance matrix of the task state sigma points ($\mathbf{P_{xx}}(t-1)$) were increased 10-fold, which, again, tuned the filter to rely more on sensory information in the following time step (i.e., $t$). The resulting Kalman gain and state correction signals became larger for the given time step as well as the following time steps, yielding large updates in articulatory-to-task transformation. As shown in Fig 4B (top), the update led to immediate adaptive response in the subsequent trial [60]. Overall, the adaptive response from AUKF was more similar to the experimental data than the simulation from the non-adaptive Kalman filter. In addition to learning, the model simulation also showed "unlearning" during the after-effects phase (i.e., after the the perturbation was removed).

The simulation also demonstrated that auditory prediction errors quickly became smaller due to auditory prediction changes (i.e., increase in predicted F1) as well as the acoustic output

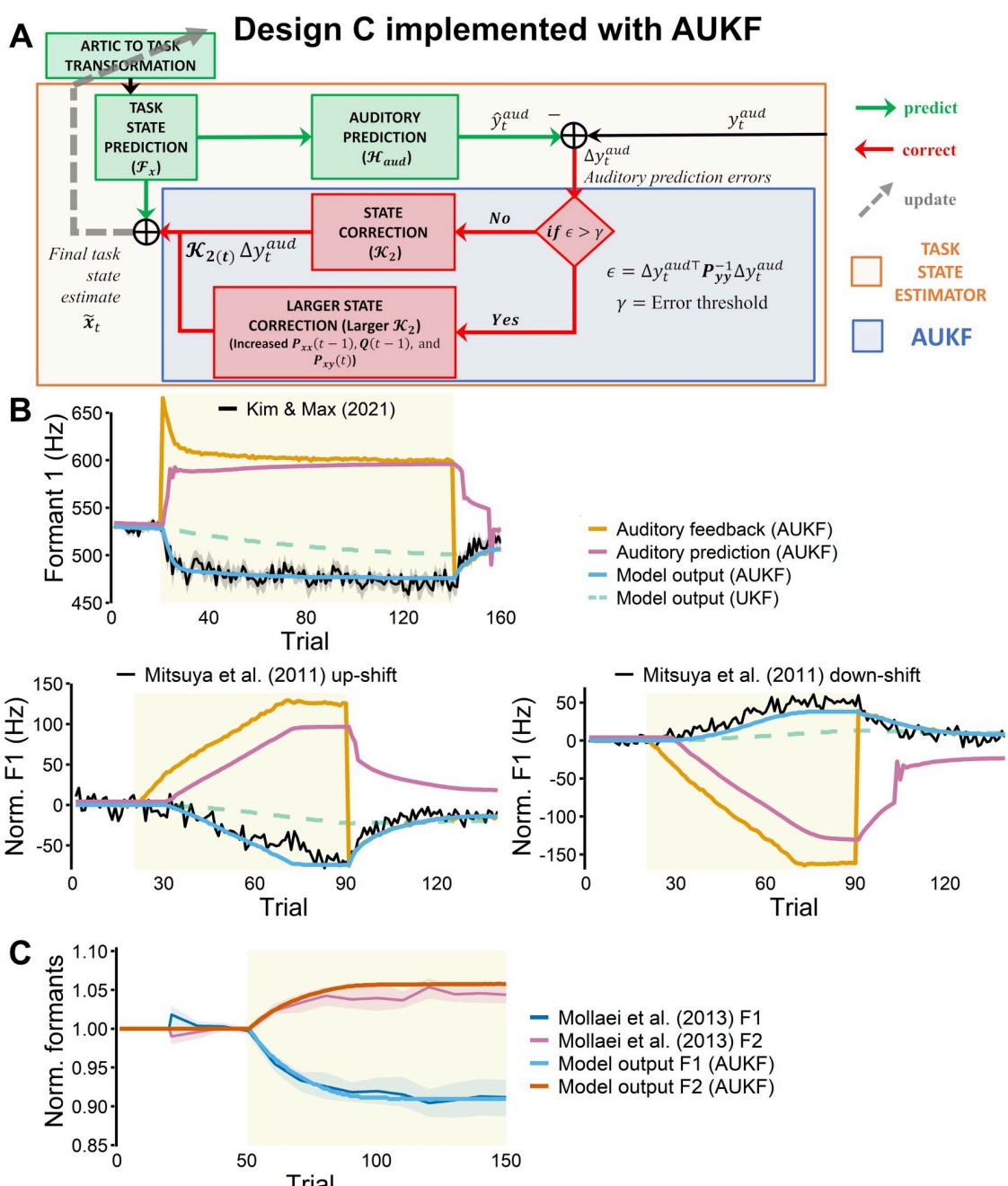

**Fig 4. Design C implemented with adaptive UKF (AUKF). A**: The task estimator (Design C) implemented with AUKF. During perturbation trials, if the squared auditory prediction errors inversely weighted by noise covariance ($\epsilon$) were larger than a given threshold ($\gamma$), larger correction signals were generated from the Kalman filter for the current and the following time step. This in turn allowed faster and larger updates in the articulatory-to-task model. **B**: Simulations are shown with data from the control group in Kim & Max [19] in the top row, and with the native English speakers in Mitsuya et al. [61] in the bottom row. In all three simulations, the AUKF (blue solid line) produced more realistic simulations compared to the non-adaptive UKF (green dashed line). **C**: The model also generated F2 changes even when only F1 was perturbed. The simulated adaptation was similar to the healthy control group's data in [62]. Perturbed trials are indicated by yellow shaded areas in both B and C.

changes (i.e., decrease in produced F1). Because the produced F1 decreased, its perturbed auditory feedback also decreased, resulting in in a smaller difference between predicted feedback and actual feedback (i.e., smaller prediction errors). In addition, the magnitude of changes in auditory prediction and acoustic output of the model were clearly coupled. When the auditory prediction errors were large (i.e., auditory prediction was very different from auditory feedback), trial-to-trial adaptive changes in the acoustic output were larger (i.e., trial 21–30). In contrast, when the auditory prediction was closer to the auditory feedback, and thus the prediction errors were smaller (i.e., after trial 40), the trial-to-trial learning was reduced even though the extent of the adaptation was largely incomplete (i.e., the perturbed auditory feedback never reached the pre-perturbation baseline output level).

The model was also able to generate realistic adaptation in response to a gradually introduced (ramped) perturbation. After tuning a few parameters (see Materials and methods for more details), the simulations closely matched the native English speakers' data from Mitsuya et al. [61] in both up-shift (i.e., F1 was increased in the auditory feedback) and down-shift (i.e., F1 was decreased in the auditory feedback) conditions (Fig 4B), bottom). Additionally, the model produced changes in the second formant frequency (F2) output even when only F1 was perturbed, similar to previous behavioral findings (e.g., [62–65]). In Fig 4C, it can be observed that the simulation closely matched both F1 and F2 adaptation response in the control group from Mollaei et al. [62], even though the model parameters were tuned only to match the F1 productions.

## Auditory prediction and articulatory state feedback control law

In order to examine the core mechanisms that drive adaptation, we ran various test simulations of the proposed Design C with adaptive UKF (AUKF). The first test examined the relationship between auditory prediction and adaptation. In the simulations of Design C (with both UKF and AUKF), auditory prediction changed in response to perturbations. Because such changes occur along with adaptation, it was desired to investigate whether such prediction change was necessary for adaptation. Therefore, we artificially fixed the auditory prediction to remain at the baseline value (i.e., no prediction changes). Despite the prediction remaining at a same value, an adaptive response was still present (Fig 5, left), showing that auditory prediction changes are not necessary for adaptation. In fact, the adaptation response was larger and faster than the default mode (dashed line) in which auditory prediction changed, suggesting that auditory prediction changes actually reduced adaptation. This observation was in line with the earlier observation that auditory prediction errors become smaller, in part, due to the auditory prediction changes. In other words, as the auditory prediction became more accurate for predicting perturbed feedback (e.g., predicting higher F1), the resulting prediction errors were reduced, thereby decreasing or slowing adaptation response.

Second, we tested the effects of updating the articulatory state feedback control law. The articulatory control law utilizes a Moore-Penrose pseudo-inverse Jacobian matrix of the articulatory-to-task transformation model (see [38, 48, 66] for more details) that transforms desired task-space changes to articulatory-space motor commands. Because the Jacobian mapping can be directly derived from any LWPR model [49], the updated articulatory-to-task mapping in Design C can be used to generate updated Jacobian matrices for the articulatory control law. We found that incorporating these updated Jacobian matrices in the articulatory control law produced a similar adaptation response to the default case, where the articulatory control law was not changed (Fig 5 right, blue solid line vs. scarlet dashed line). Importantly, when the updated Jacobian matrices in the articulatory control law were replaced with "naïve" Jacobian matrices (i.e., Jacobian matrices generated by the original, unadapted LWPR) in the middle of

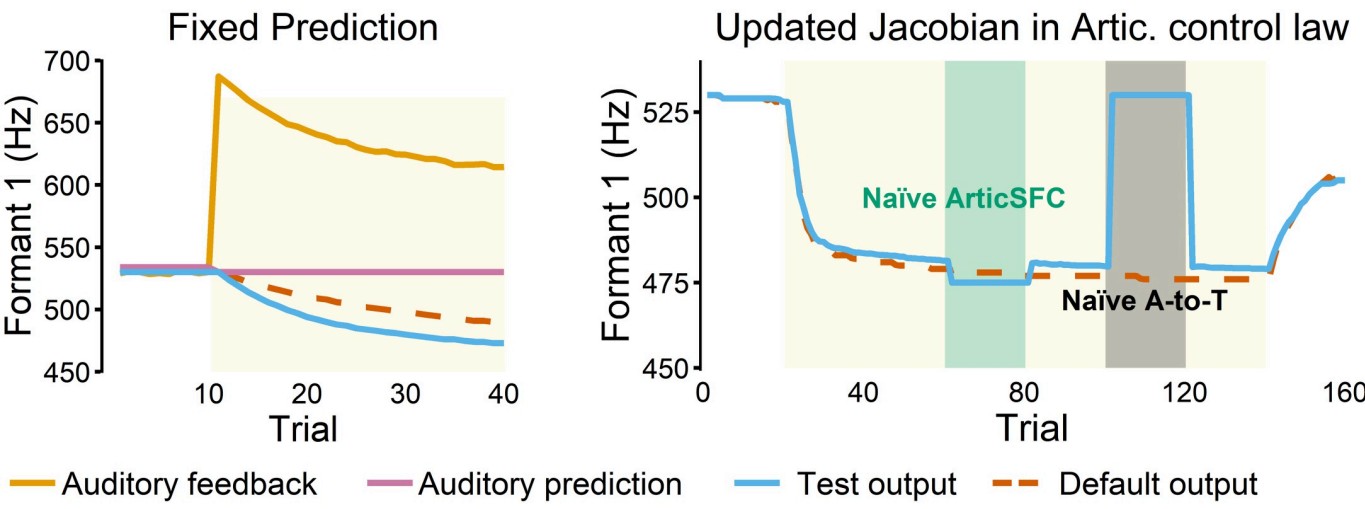

**Fig 5. Auditory prediction and articulatory state feedback control law. Left**: Simulations with auditory prediction fixed to its baseline value. The output showed a greater adaptation response compared to the default simulation where the auditory prediction was allowed to vary. **Right**: Using updated Jacobian (task-to-articulatory) matrices in the articulatory state feedback control law resulted in similar adaptation behavior (blue solid line) to that of the default mode which updates only the articulatory-to-task transformation (scarlet dashed line). When "naïve" Jacobian matrices generated from the unadapted articulatory-to-task model were used in the articulatory state feedback control law instead (green shaded area), the adaptation behavior was still present. In contrast, when the adapted Jacobian matrix for the state feedback control law was used with the "naïve" articulatory-to-task transformation model, adaptation disappeared (gray shaded area).

adaptation, the adaptive behavior was still present (green shaded area in Fig 5 right). On the other hand, when the updated Jacobean matrices were maintained in the articulatory control law while the updated articulatory-to-task transformation module was switched to the "naïve" (original) model, adaptation was completely absent (gray shaded area in Fig 5 right).

One reason for this finding may be that we used the Moore-Penrose pseudoinverse (which finds the minimum Euclidean norm solution to a system of linear equations with multiple solutions) of the Jacobian to approximate the true inverse Jacobian in the Articulatory State Feedback Control Law. Further, we found that the Jacobian matrices had surprisingly large changes in the lip parameters that appeared irrelevant for adaptation. Hence, even when the articulatory-to-task LWPR can successfully update its prediction, its updated Jacobian matrix may not capture the appropriate dimensions for the Articulatory State Feedback Control Law. An alternative possibility may be that the task state is more directly related to the acoustic output in that vowel formants are related to constrictions in the vocal tract rather than articulatory positions. In this framework (Design C), the articulatory state is most directly associated with somatosensory feedback.

## Online compensation to unpredictable formant perturbation

Although the main scope of the current study was to drive trial-to-trial sensorimotor adaptation using auditory prediction errors, such errors can also be used to generate online feedback corrections within a single utterance. Our previous model [38] with the same architecture used in Designs A and B, was able to reproduce this behavior. Here, we tested whether the revised hierarchical model (Design C) was similarly able to produce online compensation behavior for auditory feedback perturbations. In order to match previous behavioral and computational work, we lengthened vowel duration to 400 ms (80 time steps) and applied a perturbation that shifted F1 upwards by 100 Hz.

In this simulation, the default UKF setting, rather than AUKF, was used for two reasons. First, although the current model does not implement sensory feedback delays, it is intuitive that the sensorimotor system would not drastically alter ongoing productions based on the highly delayed sensory signals (see similar perspectives in the context of arm reaching in [67] or saccades in [68]). Thus, it is unlikely that the state correction signals which update the task state estimator in the context of online compensation are as large as in adaptation. In agreement with this idea, online compensation response is commonly reported to be much smaller ($\sim$10% of perturbation, [69]) than the adaptive response ($\sim$30% of perturbation [4]). Additionally, our previous work showed that a non-augmented UKF could achieve the small compensation response in the case where auditory feedback was used to estimate the articulatory state [38]. Second, recent studies have reported that the amount of online compensation and the extent of adaptation do not correlate across participants [69, 70], suggesting that a UKF governing compensation should be implemented differently from the AUKF which governs adaptation response in FACTS.

Design C replicated our previous version's online compensation response to formant shift (Fig 6 left). The size of the compensation relative to the amount of perturbation was similar to previous empirical findings (e.g., $\sim$10% in response in [69]). Although changes can be observed in both F1 and tongue height, the articulatory state estimate was closely tracking the true state (Fig 6 middle). Fig 6 (right) illustrates that the task state estimate changed abruptly immediately after the perturbation onset (i.e., time step 41–43), but quickly returned to the task target (22.1 mm), due to the articulatory state changes. Thus, the modified hierarchical FACTS architecture was also capable of generating online compensatory responses to unpredictable auditory feedback perturbation.

## Effect of model parameters on adaptation

**Adaptive UKF (AUKF) multiplicative gains ($\beta_{xx}, \beta_Q, \beta_{xy}$).** In the AUKF, the prior covariance ($\mathbf{P_{xx}}$), process noise ($\mathbf{Q}$), and cross covariance ($\mathbf{P_{xy}}$) matrices were each multiplied by gain scalars, $\beta_{xx}$, $\beta_Q$, and $\beta_{xy}$ accordingly. As shown in Fig 7A, increasing the gains increased adaptation rate and extent, though there was a larger impact on the rate than the extent. For fitting the data from Kim & Max [19], which showed a large amount of adaptation, large values for the AUKF gain were required and had to be increased by several fold above their default

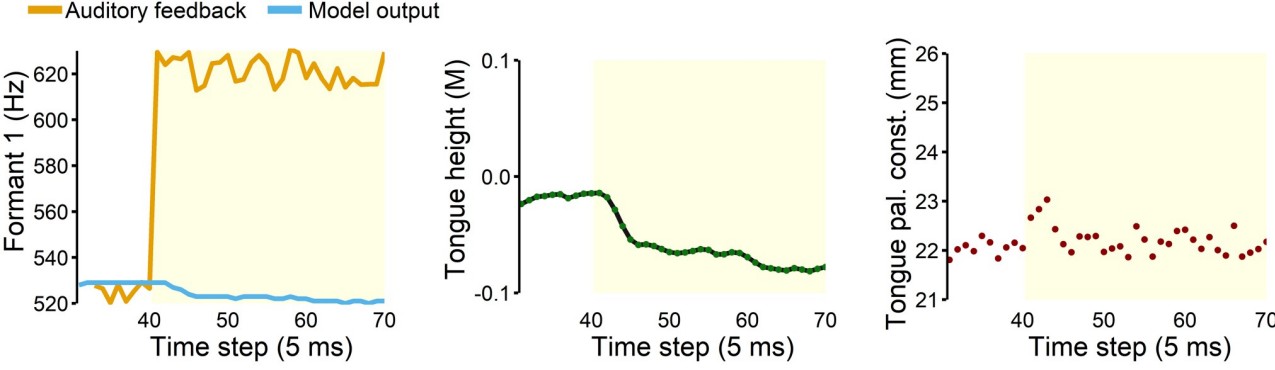

**Fig 6. Within-utterance data during online compensation response.** Yellow shaded area indicates time steps with the 100 Hz up-shift perturbation in F1. **Left**: F1 decreased across the time steps in response to the unpredictable perturbation. **Middle**: The compensatory response was also observed in the articulatory state estimate (gray dots) and its true state (green lines) expressed in the Maeda unit (M). **Right**: Despite the changes in the articulatory state, the task state estimate remained near the task target.

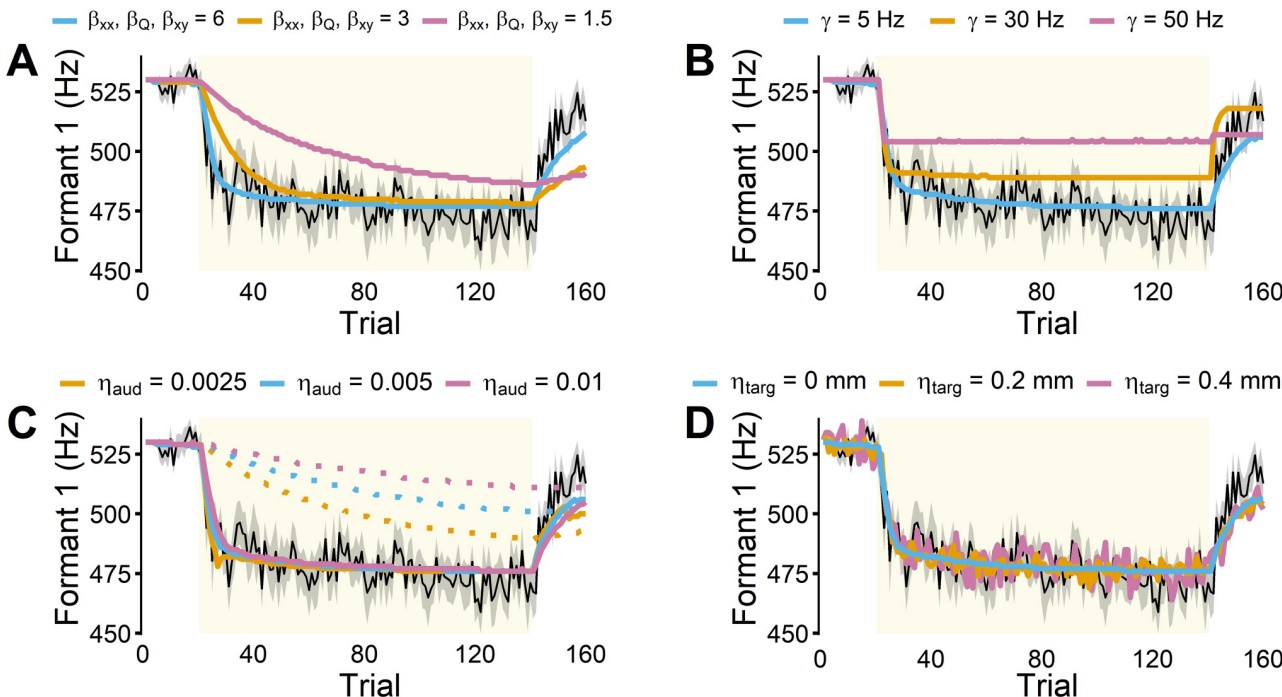

**Fig 7. The effects of changes in model parameters on adaptation.** The experimental data shown as a comparison (black line) is the control group in Kim & Max [19]. For a comprehensive overview of the parameters used in the figure, see Materials and methods. **A**: AUKF gains ($\beta_{xx}$, $\beta_Q$, $\beta_{xy}$), multiplied to the prior covariance matrix ($\mathbf{P_{xx}}$), process noise matrix $\mathbf{Q}$, and cross covariance matrix ($\mathbf{P_{xy}}$) accordingly, affected the adaptation rate and extent. Specifically, larger gains produced faster and larger adaptation. **B**: Increases in the auditory prediction error threshold ($\gamma$) reduced the extent of adaptation. **C**: Reducing the auditory noise scale ($\eta_{aud}$) increased adaptation in FACTS with UKF (dotted lines). However, changes in auditory noise scale had a minimal effect on adaptation in FACTS implemented with AUKF (solid lines). **D**: Increases in task target noise ($\eta_{targ}$) did not affect adaptive behavior, but did increase inter-trial variability.

settings. However, for fitting data in Mollaei et al. [62], lower AUKF gains provided a better fit (e.g., see the parameters used in various simulations in Materials and methods). It remains an open question how the AUKF gains may vary depending on individuals, populations, and experimental paradigms.

**Auditory prediction error detection threshold ($\gamma$).** In all the simulations presented in the current study, adaptation was driven by auditory prediction errors, consistent with theoretical proposals in the literature. To accomplish this, the final state estimate for each time step (which was updated by auditory prediction error-based state correction signals) was stored into the memory buffer to be used to update the task transformation after the utterance. This registration of error into memory occurred only if auditory prediction error in F1 for that time step was larger than a threshold (e.g., 5 Hz). Hence, the size of the error detection threshold partially determined the extent of adaptation. If the threshold was larger, the amount of adaptation was smaller because prediction errors became more rapidly smaller than the large threshold during learning, causing the model to reach a plateau (see in Fig 7B).

**Auditory noise ($\eta_{aud}$).** Measurement noise in sensory systems influences Kalman filtering as the noise covariance matrix directly determines how much the filter weighs feedback signals. FACTS is implemented with an auditory noise scalar, a parameter that determines Gaussian noise added to auditory signals [38]. As the amount of noise in the auditory signal increases, the Kalman filter relies less on auditory feedback, leading to smaller Kalman gains (i.e., smaller updates to the initial prediction). Theoretically, this would result in less

adaptation, given that the final estimate is used to update the articulatory-to-task transformation model. Congruent with this hypothesis, increasing auditory noise levels resulted in decreases in adaptation in Design C with the non-adaptive UKF (Fig 7C). In the design with AUKF, however, the effect of auditory noise had only minimal effects in terms of adaptation, likely because all auditory prediction errors above the detection threshold ($\gamma$) already drove large updates to the state in the AUKF case.

**Task target noise ($\eta_{targ}$).**   FACTS has a task target noise scalar parameter that adds Gaussian noise to the task target (i.e., constriction degree). When the task target(s) for each production/utterance were slightly altered by this added noise, the model simulated inter-trial variability. The simulations presented in the previous sections had this parameter set to 0 so that the differences between simulations could be demonstrated more clearly. When this parameter was increased, inter-trial variability increased, but the variability did not affect the overall extent of adaptation (Fig 7D). Notably, this finding that the adaptive behavior did not change even when task target for each trial was slightly altered indicated that the articulatory-to-task transformation model updates generalized to some degree beyond the specific articulatory and task state pairs used to update the model, affecting a broader region in the articulatory and task spaces.

## Discussion

We investigated sensorimotor mechanisms through which auditory prediction errors drive auditory-motor adaptation via FACTS. Whereas designs based on the original architecture of FACTS [38] did not yield adaptive behaviors, the newly modified hierarchical architecture in which auditory prediction errors updated the articulatory-to-task transformation (Design C) simulated adaptation. With a simple adaptive UKF (AUKF) implementation, Design C demonstrated realistic adaptive behaviors that were qualitatively similar to four separate experimental data sets [19, 61, 62]. The model was capable of generating adaptation in response to different perturbations including sudden (step) and gradual (ramp) as well as up-shift and down-shift F1 perturbations. The model also generated unintended adaptation in F2 during F1-only perturbations, a phenomenon that has been documented in the literature (e.g., [62]). In addition, the modified model replicated the previous version's online compensation response to unpredictable F1 perturbation.

### Auditory prediction errors

In FACTS, auditory prediction errors were shown to drive adaptation, adding further support to the perspective that auditory prediction errors may drive auditory-motor adaptation (e.g., [5, 19, 20, 26, 27]). Moreover, FACTS simulations also demonstrated that the size of auditory prediction errors directly influenced adaptive behavior. In the early perturbation trials in which there were large auditory prediction errors, the rate of learning was faster. In contrast, when auditory prediction errors became smaller, the adaptation response slowed down or reached a plateau, suggesting that the size of the prediction errors may affect the rate and extent of learning.

### Task-level updates

Importantly, adaptation was observed only when the articulatory-to-task transformation was updated, resulting in task-level changes (Fig 3). In contrast, adaptation was absent when updating only articulatory-level modules such as the articulatory state prediction (Design A) or the inverse Jacobian matrix (i.e., task-to-articulatory) in the articulatory state feedback control law (Fig 5 right, gray shaded area). Although we cannot conclude that articulatory-level

changes cannot yield adaptation in other model designs from these findings alone (see S1 Appendix for additional analyses), we show that within the existing FACTS architecture only task-level changes, and not articulatory-level changes, seem to yield adaptation.

This finding is particularly interesting in light of multiple recent upper limb motor learning studies suggesting that implicit adaptation may be influenced by task errors in addition to prediction errors [10–15]. Likewise, Ranjan & Smith [71] argued that implicit adaptation could be better explained by error signals that account for changes in both the prediction error-based estimate and the target. Thus, the authors demonstrated that the driving error signals can be described as motor performance prediction errors (MPPE), a discrepancy between the estimated action and movement goal, rather than sensory prediction errors. Directly in line with these perspectives, the task state feedback control law in FACTS computes the discrepancy between the task state estimate and the task target. As a result, in theory, FACTS can adapt from both prediction error and task error. For example, if the task estimate is different from the task target due to the updates in task estimate, the control law issues motor commands to reduce the discrepancy (e.g., Fig 4). Similarly, if the task target is shifted, the control law would also issue corrective motor commands for the errors to be reduced (i.e, moving task estimate towards the shifted target). Hence, the FACTS design offers a plausible hierarchical state feedback control framework in which both prediction errors and task errors can contribute to adaptation. This is especially important given that studies continue to examine how task errors or task-space variables may impact speech adaptation (e.g., anchoring sound target, [72], lexical status of the target, [73]).

Adaptation led by task-level changes has been previously reported by other prominent computational models. Most notably, the DIVA [29] and GEPPETO [58, 74] models both demonstrated that updates in the internal models that govern task space, which was defined as the auditory space in the models, led to adaptation. Even though the task space in FACTS is defined in terms of constrictions, our results replicated the same result. In sum, the importance of task-level updates is highlighted in the FACTS simulations, congruent with both speech and non-speech motor control literature.

## Articulatory-to-task transformation updates

The articulatory-to-task transformation module is novel and its mechanism in driving adaptation can be interpreted with different perspectives. The first perspective is that the task transformation updates may reflect forward model updates. By its design, task transformation provides input to the task state prediction, so any updates in the transformation greatly influences the task prediction and the resulting task state estimate. In this regard, task transformation updates can be seen as a type of forward model-based learning that updates the task estimate, which in turn causes the task state feedback control law to generate adaptive behavior. Indeed, our finding that auditory forward prediction changed during adaptation supports this notion. This perspective is also similar to the idea of paired inverse-forward model updates (e.g., [34]), and has been demonstrated to be plausible in a simple formant adaptation model [37]. Nevertheless, strictly speaking, the paired inverse-forward model updates are not congruent with FACTS given that the task state feedback control law is not an inverse model of the task transformation (for similar views on control policies vs. inverse models, see [35, 42]).

An alternative, perhaps more straightforward, perspective is that the task transformation updates reflect task-level control changes, rather than changes to forward models. This interpretation arises from the view that task-level control changes, but not prediction updates, are ultimately required to drive learning [36]. For example, our simulations illustrated that forward model updates alone did not drive adaptation, even though the auditory prediction was

successfully updated to predict the perturbed feedback (see Design B in Fig 3). Indeed, updating only the auditory prediction to match the perturbed auditory feedback could not possibly result in adaptation because the error signals would be reduced without any changes in the motor behavior (e.g., [75]). In contrast, even when auditory prediction was artificially fixed at the baseline level, adaptation response was present (Fig 5 left) as long as the task-level control was updated. Thus, within the FACTS framework, the forward model-based updates were not necessary for adaptation. Rather, the articulatory-to-task transformation updates, which can be seen as direct updates to task-level control, drove adaptation.

Importantly, the alternative perspective does not necessarily reject the notion that forward model updates may be present during adaptation. That is, the task transformation inevitably affects auditory prediction (in addition to the task state feedback control law), and it may be viewed also as a forward model (in addition to being a part of the control law) as in the first perspective. However, in contrast to the first perspective, the alternative perspective suggests that forward model updates, though they may be present, do not drive the learning behavior (see a similar view in [36]).

## Incomplete adaptation

The FACTS simulations replicated previous findings of incomplete speech adaptation, which is generally reported to be 20%-40% of the perturbation magnitude [4]. Some have hypothesized that the mismatch between auditory and somatosensory feedback may lead to incomplete adaptation (e.g., [76–78]). The current version of the FACTS model (Fig 2), unlike the previous version [38], does not directly integrate auditory and somatosensory feedback-based state corrections, and thus cannot provide support for this view.

Others have suggested that speech adaptation may have small error sensitivity, a broad term in state space models that describes how much of the perceived error translates to learning (e.g., [79]). In FACTS, there are multiple model parameters that can affect the model's sensitivity to errors. However, even when some of those parameters were set to be highly sensitive to auditory errors (i.e., auditory prediction error threshold < 5 Hz or auditory noise scale effectively < 2–3 Hz), the adaptation response remained short of 60 Hz or greater compared to full adaptation. This suggests that the model parameters that determine sensitivity to auditory errors were not the main factors that limited adaptation in FACTS simulations.

It also has been suggested that the physiological constraints of the vocal tract may be limiting the amount of adaptation (e.g., [4]). However, the FACTS vocal tract, the Maeda model, is capable of lowering F1 to reach full adaptation (see the vowel space described in [80] and S2 Appendix). Thus, the incomplete adaptation in FACTS cannot be explained by the limitation in the model. Another possibility is that changing perceptual boundaries during adaptation may be shifting auditory targets, limiting adaptation [81]. However, recent studies have found that perceptual targets do not change during adaptation [19] and providing accurate perceptual targets to participants (by playing the target sounds to the participants) does not increase adaptation [72, 82].

Instead, FACTS simulations provide another plausible explanation for incomplete adaptation: auditory prediction. During adaptation, the F1 auditory prediction increased across the trials in order to better predict the perturbed feedback (i.e., increased F1), but the actual auditory feedback decreased due to the adaptive changes in the motor output (Fig 4B). As auditory prediction and auditory feedback moved towards each other, the resulting auditory prediction errors were drastically reduced across the perturbation trials. As a result, by the time auditory prediction errors were no longer registered as errors (i.e., became smaller than the auditory prediction error threshold) and the adaptation halted, the extent of adaptation remained

incomplete. Thus, adaptation extent was limited by changes in auditory prediction (in addition to other factors such as the error threshold, see below). This factor contributing to incomplete adaptation may also be applicable to other models of speech motor control such as Bayesian GEPPETO, in which the auditory-motor internal forward model is updated during learning [58, 74]. Hence, this idea warrants future empirical studies.

## Adaptive changes in F2

FACTS generated adaptive changes in both F1 and F2, even though the perturbation was applied only in F1. Such F2 increase in response to the F1 up-shift perturbation has been documented by multiple studies [62–65], but also see contrary findings in [76, 83]). Furthermore, there is evidence that such F1 and F2 changes may be coupled together. Lametti et al. [64] reported that anodal transcranial direct current stimulation (tDCS) enhanced adaptation in both F1 and F2, even though only F1 was perturbed. Tang et al. [65] corroborated the finding by demonstrating that a repetitive transcranial magnetic stimulation (rTMS) application over the tongue region of M1 impaired adaptation response in both F1 and F2.

It has been postulated that such changes in F2 might be due to the speech motor system choosing to issue motor commands that correspond to an already learned speech production (e.g., [65]). For example, the sensorimotor system may attempt to lower F1 by changing its productions closer to another vowel, /I/, which may increase F2. While this may be true, the FACTS model is not implemented with any preference in the task or articulatory space. Instead, from our model design, we propose two additional potential explanations for this phenomenon. First, this may be due to the fact that the tasks are defined as constrictions, and therefore, task-level constriction-based changes during adaptation may result in changes in both formant frequencies. That is, the task state feedback control law reducing the tongue palatal constriction degree not only lowers F1, but also increases F2. Second, there may be biomechanical reasons why changing the tongue position to counter the F1 shift also changes F2. In fact, studies of sensitivity functions in the vocal tract show that any change in vocal tract area function induces simultaneous changes in F1, F2 and F3 (e.g., [84]). However, this latter point is not supported by our previous study with the Maeda model that showed a nearly full coverage of the vowel space in the Maeda model output ([80] and also see S2 Appendix). This means that, at least for the target vowel used in the current study (/ɛ/), the adaptive behavior in FACTS should not be constrained by the kinematics of the Maeda model (i.e., articulatory-level changes can certainly be made to alter F1, but not F2).

## Perceptual acuity and auditory noise

Previous research has examined the relationship between perceptual acuity and adaptation, and found that perceptual acuity could predict the amount of F1 adaptation (e.g., [85, 86]). Likewise, a recent study by Daliri & Dittman [26] also found that perceptual boundary sizes, measured from psychometric functions, negatively correlated with adaptation (i.e., participants with smaller perceptual target boundary size adapted more). A computational model, DIVA, has also demonstrated that the auditory acuity defined as the size of auditory targets is negatively correlated with adaptation [85]. In FACTS, the perceptual auditory acuity may be most directly related to two parameters. First, auditory noise has been hypothesized to be inversely related to the perceptual acuity [38]. Interestingly, simulations regarding the effects of auditory noise on adaptation told conflicting stories. The auditory noise clearly affected adaptation in the design with non-adaptive UKF, congruent with previously mentioned empirical findings. On the other hand, in a simulation with AUKF that yielded faster and larger adaptation, matching experimental data, the effects on the rate and amount of

adaptation were nearly negligible, likely because all auditory prediction errors above the detection threshold already drove most of the learning.

The other model parameter potentially related to the perceptual acuity is auditory prediction error threshold. In FACTS, auditory prediction error threshold showed a clear effect on adaptation. It is difficult, however, to interpret our findings because the documented empirical findings regarding the effect of perceptual acuity on adaptation are largely inconsistent. Even though some have found a link between perceptual acuity (or auditory error detection abilities) and adaptation extent [26, 85, 86] as mentioned above, several other studies also have reported that F1 acuity was not associated with the amount of F1 adaptation [87, 88]. Likewise, pitch acuity was found to be unrelated to pitch adaptation in studies [89, 90]. In addition, Nault & Munhall [86] also did not find a correlation between perceptual acuity and adaptation in F2 (unlike F1). Moreover, although Martin et al. [91] reported that two measures of perceptual acuity could predict the amount of formant adaptation, the perceptual acuity measures were from the loudness discrimination and pitch discrimination tasks, rather than formant-related measures.

In addition, the implicit nature of speech auditory-motor adaptation [19] raises a question as to whether the methods used for measuring perceptual acuity can accurately probe sensorimotor mechanisms involved in auditory prediction error detection. Perceptual acuity is measured explicitly, by the psychometric measures or Just Noticeable Difference (JND) tasks which prompt participants to report whether they heard a difference between sounds or not. Although explicit report does not necessarily mean that what is being reported was the result of an explicit process, it is possible that such acuity measure may fundamentally differ from the actual sensorimotor mechanisms involved in adaptation, given that participants can often adapt even when they are not explicitly aware of the perturbation [16, 92]. Taken together, the relationship between the perceptual acuity and adaptation as well as the relevance of these model parameters to perceptual acuity and adaptation remain unclear at this time.

## Inter-trial variability

The large inter-trial variability in speech production have intrigued many researchers. In the current FACTS design, although the speech plant noise in the articulators (representing neuromuscular noise) can induce some time step-to-time step variability, such inter-time step variability does not contribute much to inter-trial variability because each trial production is computed as an average of multiple time steps. Thus, the only source of inter-trial variability is the task target noise, an account that remains to be investigated in future studies. Nevertheless, there is some evidence that is in line with FACTS simulations of inter-trial variability.

First, jaw or tongue movement end points for a given vowel target can vary by several millimeters across trials, even during natural un-perturbed speech movements generated in a largely feedforward manner (e.g., [93]). In other words, there is a considerable amount of inter-trial variability even when movements are likely reaching their targets, suggesting that the idea of task target noise is plausible. Second, there is evidence that inter-trial variability may be controlled by the same mechanism during both baseline trials and perturbation trials [86]. In the study, the authors reported that individuals with high baseline formant variability also had high formant variability during perturbation trials and the amount of variability in the two phases did not differ. These accounts are in accordance with FACTS in which a) inter-trial formant variability is controlled by a single model parameter, task target noise, and b) the variability remains the same across baseline and perturbation trials. Third, inter-trial variability does not seem to be related to the amount of adaptation (Fig 7D). Although Purcell & Munhall [94] reported a significant correlation between inter-trial variability and adaptation, the

relevance of the finding has been questioned due to an overdifferencing issue in their methodology (see [95] for more details). Likewise, Nault & Munhall [86] found a significant correlation between baseline inter-trial variability and adaptation, but the authors also stated that their finding might be due to undersampling (among other issues), adding that a larger data set from seven experiments did not find such correlation. Furthermore, Wang & Max [95] recently demonstrated that reducing or magnifying the perceived inter-trial variability during the baseline (i.e., before the perturbation trials) changed the overall production variability in the baseline production, but such changes in perceived and production variability did not affect the adaptation performances. Hence, our simulations in which inter-trial variability does not affect adaptation are in agreement with the literature.

Two recent studies found that artificially altering perceived inter-trial formant variability in the auditory feedback resulted in compensatory changes in formant production variability, suggesting that inter-trial variability is actively controlled and regulated [95, 96]. In the current FACTS design, the task target noise variable is defined by a configuration file rather than being actively controlled during simulations. Nevertheless, FACTS already incorporates sensory noise covariance matrices, so it remains possible to incorporate such mechanism for tracking and controlling inter-trial production variability.

It should be noted that other computational models of speech implement target noise as well. One example is GEPPETO, the optimal motor control model that controls a biomechanical model of the tongue, was recently re-developed within a Bayesian framework [97]. In the model, motor control variables were randomly selected under a probability distribution, introducing inter-trial variability, a concept analogous to the target noise in the FACTS framework. In DIVA, auditory targets are defined as time-varying regions, which allow for articulatory variability within these regions [29].

## Unlearning

Although learning behavior was the main scope of this investigation, it should be noted that the model also generated unlearning behavior in the trials after the perturbation was removed (i.e., aftereffects). Importantly, such unlearning behavior was also driven by auditory prediction errors in a manner similar to learning. That is, during the aftereffects phase, the unlearning was faster when auditory prediction and auditory feedback were very different. However, as auditory prediction and feedback moved closer to one another across the trials, the unlearning behavior quickly slowed down, suggesting that unlearning may also be driven by auditory prediction errors.

## Neural substrates of adaptation

It is generally accepted that the cerebellum plays a role in computing sensory prediction errors (e.g., [75]). Much of the supporting evidence comes from the fact that people with cerebellar diseases show poor adaptation across many effector systems (e.g., gait, [98]; reaching, [32]; speech, [27]). Though applying anodal transcranial direct current stimulation to the cerebellum showed inconsistent effects on reaching adaptation [99–101], one study has found its enhancing effect on speech adaptation [64]. In addition, cerebellar activity changes associated with adaptation have also been found (see [102] for review). These lines of evidence are in line with our model that simulates prediction error-driven adaptation.

In addition to the cerebellum, multiple cortical structures are activated during nonspeech adaptation [102]. In particular, previous studies found that the posterior parietal cortex (PPC), which has been associated with task state estimate (or task state estimator), may change during adaptation (see [53] for a review). Interestingly, several areas of PPC have been found to be

associated with speech auditory-motor adaptation as well. Shum et al. [103] found that applying repetitive transcranial magnetic stimulation (rTMS) over the suprmarginal gyrus (SMG) in the left inferior parietal lobe (IPL) reduced the extent of formant adaptation. Another study reported that transcranial direct current stimulation (tDCS) to IPL enhanced formant adaptation, even though cathodal tDCS did not reduce adaptation potentially due to tDCS being less focal than TMS [104]. Crucially, the connectivity between PPC areas (e.g., the left IPL) and auditory areas (e.g., anterior superior temporal sulcus) are altered following F1 adaptation [16]. Hence, these lines of evidence are consistent with the present study in that task-level changes are associated with adaptation.

The premotor cortex (PM) has also been implicated with auditory-motor adaptation. Multiple functional network connectivity involving the right inferior frontal gyrus (IFG) were found to be correlated with adaptation extent [105]. Floegel et al. [16] also reported that the resting-state functional connectivity between the right superior temporal sulcus and the right IFG was changed after auditory-motor adaptation. Similarly, functional connectivity between right IFG and right PPC areas, namely posterior SMG and anterior angular gyrus, was found to be positively correlated with adaptation performances [105]. Given that the ventral PM is thought to be the location for the task state feedback control law ([38] or "task action" in [53]), these accounts are also in agreement with our simulations in which task-level changes in the task state feedback control law (i.e., issuing modified task motor commands) lead to adaptation.

Recent studies have also emphasized the crucial role of the primary motor cortex (M1) in auditory-motor learning. The M1 ɣ-aminobutyric acid (GABA) level was found to be linked with the amount of auditory-motor learning of controlling joystick in directions associated with sounds [106]. In addition, Tang and colleagues [65] found that formant adaptation was absent after applying rTMS to the tongue region in M1. Moreover, F1 adaptation was enhanced by anodal tDCS of M1 [64, 83]. Given that M1 is known to encode task-specific muscle synergies in speech [107], it is possible that it may also play a role in the articulatory-to-task transformation updates. Taken together, though largely speculative, these narratives demonstrate that the FACTS design and simulation may be neurally plausible.

## Limitations and future directions

In this study, we described the first hierarchical state feedback control model that simulates speech adaptation. Given the exploratory nature, however, there are multiple limitations in the model. One apparent limitation is that the amount of simulated adaptation scales with the amount of perturbation. This is contrary to previous findings showing that the amount of adaptation can plateau [76] or even decrease in proportion [76, 77, 108] in response to a very large perturbation. Interestingly, even the amount of online compensation also proportionally decreases as the perturbation size increases [109], suggesting that there may be a broad mechanism that determines the validity of auditory signals, specifically examining the likelihood of auditory feedback being internally generated as opposed to externally generated. In a future study, this functionality may be modeled by adding a probability function in the computations of auditory prediction errors in the task state estimator.

Additionally, all simulations presented in the study involved perturbations to only F1. However, previous studies have shown that adaptation also occurs when perturbation is in both F1 and F2 (e.g., [7]), or only in F2 (e.g., [110]). Indeed, adaptation response can be found when F1 and F2 are perturbed in the same direction [7, 111] or in the opposite directions [76,

112]. Therefore, future studies are warranted to examine the model in different adaptation tasks.

The current version of FACTS also lacks laryngeal control and does not produce pitch. Adaptation to pitch perturbation is also thought to be implicit (e.g., [18]) and FACTS can be improved to simulate pitch adaptation that is driven by prediction errors. Previously, a state feedback control model was developed to control one dimension variable which, theoretically, can represent pitch frequency [113]. Recently, Weerathunge and colleagues [114] combined the DIVA model with a vocal fold model to develop a neurocomputational model for laryngeal motor control (LaDIVA). Their model simulated realistic compensation as well as adaptation response, providing an excellent guide for implementing laryngeal control in FACTS.

In the current study, we used LWPR for all learnable modules such as forward models and articulatory-to-task transformation. However, it remains possible that implementing alternative models such as Deep Neural Network (DNN) could result in different findings. For example, a more robust way to invert the updated articulatory-to-task transformation may yield more adaptation. Future studies should test these possibilities.

Lastly, among the previously mentioned features yet to be implemented in FACTS [38], sensory delays remain particularly important for several reasons. First, it has long been known that speech production is sensitive to (artificially) delayed auditory feedback (e.g., [115]). In fact, adding artificial delays in the auditory feedback reduces [116] or even completely eliminates adaptation response [21, 22]. Second, implementing sensory delays could make FACTS settings become more universal across different contexts. In the current study, certain mechanisms/parameters had to be explicitly selected (e.g., UKF rather than AUKF and increased auditory noise) to achieve small online compensation response. A future version of FACTS might minimally rely on online sensory feedback because sensory information is inherently delayed, which would lead to small online compensation responses. Third, sensory delays are likely to play a critical role in multi-sensory integration [67, 117, 118] and implementing sensory delays in the model would lead to a more accurate model that can simulate the complexities of multi-sensory integration in speech motor control [38]. Fourth, implementing feedback delays would serve as the first step towards simulating adaptation induced by temporal perturbation (e.g., [16, 119]).

Despite the limitations, the proposed design paves the way for a hierarchical state feedback control framework to be examined in the context of adaptation. As an initial attempt, our model demonstrated that 1) auditory prediction errors can drive speech auditory-motor adaptation through task-level updates, and 2) adaptation is likely driven by changes to task-level control rather than (only) to forward predictive models. Additionally, simulating adaptation with FACTS generated a number of important hypotheses such as: incomplete auditory-motor adaptation might be due to auditory prediction updates and not conflicts with somatosensation, and changes in F2 during adaptation to F1 perturbation may be due to the updates in the task-level representations. These hypotheses should be tested in future studies. Importantly, given that our model design is closely aligned with the non-speech motor control literature (e.g., [53]), the suggested framework and architecture may be applicable to future investigations of non-speech effector systems as well.

## Materials and methods

The current version of FACTS was written in Python 3 [120], specifically with the purpose of being open-source, extendable, and easy to redistribute. All modules and libraries that were originally developed by other researchers and developers were either re-written or wrapped in

Python as well. The source code and simulation configurations presented in the study can be accessed on a public GitHub repository (https://github.com/kwangsk/FACTS). To promote reproducibility, a YAML file included with the repository can be used to build an Anaconda virtual environment containing all package dependencies necessary to reproduce the results presented in the current study.

## Notation

In this paper, state variables are denoted with accents, namely the dot (e.g., $\dot{x}$), tilde (e.g., $\tilde{x}$), and hat (e.g., $\hat{x}$), which respectively denote derivatives, estimates, and forward predictions (generated via LWPR). Bold and capital notation describe matrix data structures (e.g., $\mathbf{Q}$) and LWPR models ($\mathcal{F}_a$). Bold and lower case notation describe state variables with more than dimension (e.g., $\tilde{\mathbf{x}}_t$ containing both position and velocity). Specifically, a higher-level task representation (vocal tract constriction degrees) is denoted with $\mathbf{x}$ (e.g., $\mathbf{x_t}$) and a lower-level articulatory representation is denoted with $\mathbf{a}$ (e.g., $\mathbf{a_t}$). Lastly, Regular (unbold) notation describe scalars (e.g., $\beta_{xx}$) or array data structures with a single dimension (e.g., task position $\tilde{x}_t$). Please see Table 1 for a comprehensive list of notation.

## Task state

From the data presented in our previous study [80], we found that a discrete number of constriction locations could produce a wide range of F1 and F2. Therefore, in the current version of FACTS, the task parameters are defined as constriction degrees (i.e., distance between the tongue and the vocal tract wall/palate) at five fixed constriction locations (Fig 8). Of the seven task parameters, the first five pertain to the tongue, namely tongue tip dental (TT_Den), tongue tip alveolar (TT_Alv), tongue body palatal (TB_Pal), tongue body velar (TB_Vel), and tongue body pharyngeal (TB_Pha). The last two task parameters pertain to the lips, namely lip aperture (LA) and lip protrusion (LPRO) magnitudes.

The task state ($\tilde{\mathbf{x}}_t$) includes both position ($\tilde{x}_t$) and velocity ($\dot{\tilde{x}}_t$) of the seven constriction degrees:

$$\tilde{\mathbf{x}}_t = \begin{pmatrix} \tilde{x}_t \\ \dot{\tilde{x}}_t \end{pmatrix} \tag{1}$$

The same composition (i.e., seven position and seven velocity) also applies to the task state prediction ($\hat{\mathbf{x}}_t$).

## Articulatory state

The variables that comprise the articulatory state vector ($\mathbf{a}_t$) in this paper differ from the previous report [38] because the speech production plant was changed from the CASY model (Configurable Articulator Synthesizer, [121]) to the Maeda model ([122] and also see S2 Appendix). The Maeda model is a data-driven model of which each input parameter is defined to be a Principal Component of articulators' positions retrieved from x-ray images. The model has been widely used, including in the well known DIVA model [29] as well as in various studies involving computational simulations (e.g., [54]). It should be noted that the choice of the plant does not affect our conclusions since FACTS is modularly designed to be able to work with different plant models and to generate equivalent simulations.

The articulatory state in the current version of the model is, therefore, defined as: jaw height (JA), tongue height (TG), tongue shape (TS), tongue apex (TA), lip height (LH), and lip protrusion (LP). These variables are expressed in the units of the Maeda Principal Component,

**Table 1. Notation for FACTS.** Bold and capital notation describe matrix data structures (e.g., $\mathbf{Q}$) and LWPR models ($\mathcal{F}_a$). Regular (unbold) notation describe scalars (e.g., $\beta_{xx}$) or array data structures with a single dimension (e.g., task position $\tilde{x}_t$).

| Notation | General FACTS variables |
|---|---|
| $\mathbf{x}$ | Task state based on vocal tract constriction degrees (both position and velocity) |
| $\mathbf{a}$ | Articulatory state describing articulatory position and velocity |
| $\eta$ | Gaussian random noise added to various variables such as task target, plant, and sensory feedback |
| $y^{somato}$ | Somatosensory feedback |
| $y^{aud}$ | Auditory feedback |
| $\Delta y^{somato}$ | Somatosensory prediction error |
| $\Delta y^{aud}$ | Auditory prediction error |
| $t$ | Time step |
| $z^{-1}$ | Time step delay operator |
| $\mathcal{F}_\mathbf{a}$ | Forward prediction model for the articulatory state |
| $\mathcal{F}_\mathbf{x}$ | Forward prediction model for the task state |
| $\mathcal{H}_\mathbf{somato}$ | Forward prediction model for the somatosensory feedback |
| $\mathcal{H}_\mathbf{aud}$ | Forward prediction model for the auditory feedback |
| $\mathbf{M}$ | Mass matrix in the task state feedback control law |
| $\mathbf{B}$ | Damping matrix in the task state feedback control law |
| $\mathbf{K}$ | Stiffness matrix in the task state feedback control law |
| $G_N$ | Gating vector for the neutral attractor |
| $B_N$ | Damping vector in the neutral attractor |
| $K_N$ | Stiffness vector in the neutral attractor |
| | |
| | **Unscented Kalman Filter variables** |
| $\mathcal{K}$ | Kalman gain for the state correction. |
| $\mathbf{P_{xx}}$ | Prior covariance matrix for the task state sigma points |
| $\mathbf{Q}$ | Process noise scale, which is added in the prior covariance matrix |
| $\mathbf{P_{xy}}$ | Cross covariance of the task state and sensory (auditory) feedback sigma points |
| $\mathbf{P_{yy}}$ | Covariance of the auditory feedback sigma points |
| | |
| | **Adaptive Unscented Kalman Filter (AUKF) variables** |
| $\epsilon$ | Square of the auditory prediction errors, inversely weighted by the auditory noise covariance matrix ($\mathbf{P_{yy}}$) |
| $\gamma$ | Error threshold for AUKF. Only when $\epsilon$ is larger than this threshold, AUKF is activated |
| $\beta_{xx}$ | Multiplicative gain for the prior covariance noise matrix of the next time step ($\mathbf{P}_{xx}(t+1)$) when AUKF is activated |
| $\beta_Q$ | Multiplicative gain for the process noise matrix of the next time step ($\mathbf{Q}(t+1)$) when AUKF is activated |
| $\beta_{xy}$ | Multiplicative gain for the cross covariance noise matrix of the current time step ($\mathbf{P}_{xy}(t)$) when AUKF is activated |
| | |
| | **Simulation variables** |
| F1 | First formant frequency |
| F2 | Second formant frequency |
| F3 | Third formant frequency |

which we refer as M in this paper. The common range for M is from −3 to 3, which can produce most sounds across the vowel space [80]. The Maeda model has one more parameter called Larynx, which controls the overall length of the larynx but the effect of the parameter on F1 and F2 was relatively minimal, so it was excluded from our state variable and its neutral

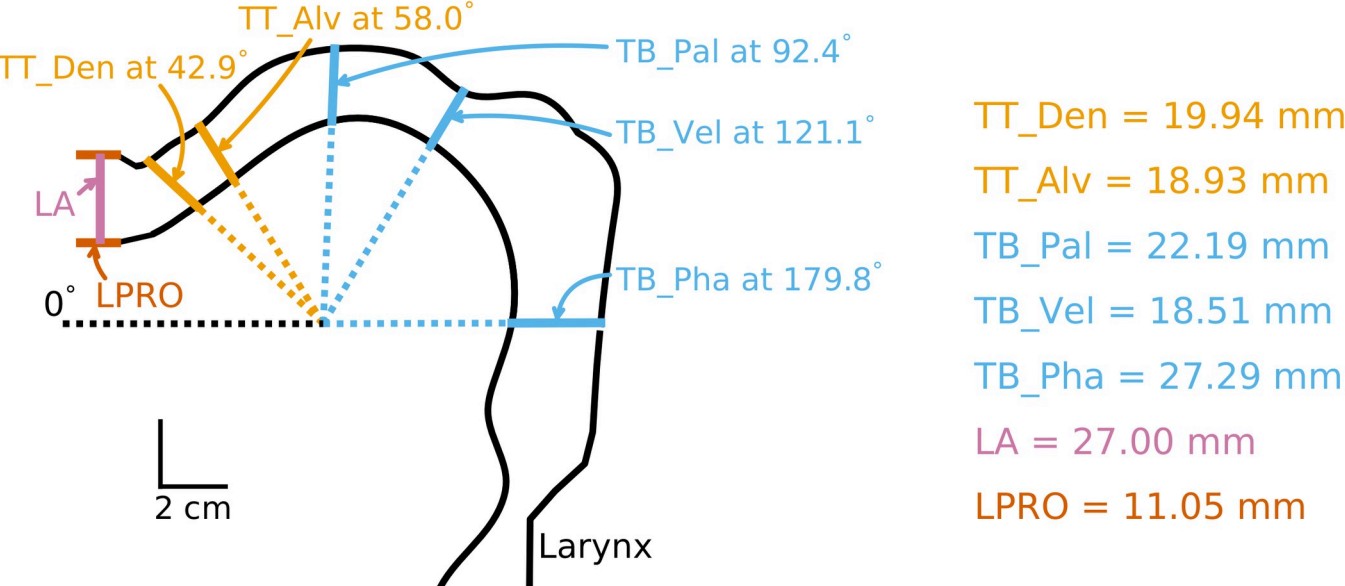

**Fig 8. Task state variables.** Of the five tongue variables, the first two pertaining to tongue tip are defined as tongue tip dental (TT_Den) and tongue tip alveolar (TT_Alv) constriction degrees, each defined at 42.9˚ and 58.0˚ from the left part of the horizontal axis (0˚). The tongue body variables are tongue body palatal (TB_Pal), tongue body velar (TB_Vel), and tongue body pharyngeal (TB_Pha) which are defined at 92.4˚, 121.1˚, and 179.8˚ respectively. The lip aperture (LA) is defined as the distance between the upper lip and the lower lip. The lip protrusion (LPRO) is the horizontal length of the upper (or lower) lip. Here, the task state of the initial position used for each trial is depicted as an example.

position (i.e., 0) was given to the Maeda model [80]. As a result, the articulatory state in FACTS consists of six position Maeda parameters and their associated velocities. The articulatory state at a time step (t) was defined as:

$$\mathbf{a_t} = \begin{pmatrix} a_t \\ \dot{a}_t \end{pmatrix} \qquad (2)$$

Although not explicitly described here, the articulatory state estimate ($\tilde{\mathbf{a}}_t$) and the articulatory prediction ($\hat{\mathbf{a}}_t$) are also defined with the same composition (six position and six velocity state variables).

## Modules

The principles of object-oriented programming were applied to create a modular code design ideal for testing multiple FACTS architectures and implementations. The Model class (*Model. py*) is responsible for dynamically building a model based on a chosen architecture. The model architectures compared in this work, for example, are implemented as a series of child classes inheriting from the parent class (*Model*). A Model object is composed of a set of modules, which correspond to the subfunctions of the theoretical FACTS model. Each module is also built with inheritance structure so that different implementations of each module can be easily tested and compared. The main script (*FACTS.py*) constructs the Model object and its modules based on the settings defined in a configuration file formatted in INI, and calls the object's run method in a nested loop to generate simulations across time steps and trials. This code design is easily extensible because new architectures and module implementations can be added without disrupting the existing code.

**Gestural score (*TADA.py*).** As in the previous FACTS, the gestural score module is a simplified version of several components from the Task Dynamics Application model (TADA

[66]). The format of gestural scores (e.g., *KimetalAdapt.G* for adaptation simulations) includes the task variable (TB_Pal), onset time step (0), offset time step (40), task target (22.1 mm), stiffness coefficient (14), and damping coefficient (1). The file also includes information on articulatory weighting and other TADA parameters although they are not currently implemented in FACTS. By default, *TADA.py*, which reads the gestural score files, is configured to assume that each time step is defined as 5 ms. The module also has the capability to add Gaussian noise to the target specified in the gestural score, yielding a slightly altered task target ($x_{target}$).

**Task state feedback control law (*TaskSFCLaw.py*).** The task state feedback control law is based on the Task Dynamics model developed by Saltzman & Munhall [48], and is defined as:

$$\ddot{x}_t = \mathbf{M}^{-1}(-\mathbf{B}\dot{\tilde{x}}_t - \mathbf{K}(\tilde{x}_t - x_{target})) \tag{3}$$

where the mass matrix (M), damping coefficient matrix (B), and stiffness coefficient matrix (K) of the second-order dynamical system model are determined based on the corresponding values configured in the gestural score file. The computed task acceleration $\ddot{x}_t$ which we refer as task motor command is sent to the articulatory state feedback control law. $x_{target}$ is the task target for time step $t$, defined by *TADA.py* (in the simulations, the target remained constant across all time steps and trials). In the modified hierarchical architecture (Design C), the task motor command's copy is also sent to the task state estimator (i.e., task efference copy).

**Articulatory state feedback control law (*ArticSFCLaw.py*).** The articulatory state feedback control law, based on the Forward Task Dynamics model, computes $\ddot{a}_t$ which is the articulatory acceleration or articulatory motor command [48]:

$$\ddot{a}_t = \mathbf{J}(\tilde{a}_t)^\dagger \ddot{x}_t - \mathbf{J}(\tilde{a}_t)^\dagger \dot{\mathbf{J}}(\tilde{a}_t, \dot{\tilde{a}}_t)\dot{\tilde{a}}_t + (\mathbf{I}_n - \mathbf{J}(\tilde{a}_t)^\dagger \mathbf{J}(\tilde{a}_t))\ddot{a}_d + \ddot{a}_N \tag{4}$$

In this equation, $\mathbf{J}(\tilde{a}_t)$ is the Jacobian Matrix (articulatory-to-task state) given the articulatory state estimate, which is computed from the naïve task transformation LWPR (i.e, not learned, but updating this Jacobian matrix has minimal effect on adaptation as shown in Fig 5B). Effectively, the Moore-Penrose pseudoinverse of the Jacobian matrix is denoted as $\mathbf{J}(\tilde{a}_t)^\dagger$. The true derivative of the Jacobian over time is expressed as $\dot{\mathbf{J}}(\tilde{a}_t, \dot{\tilde{a}}_t)$. In our design, the derivative of the Jacobian is not readily available as it is in TADA, because the relationship between articulatory variables to task variables is not determined from a set of mathematical (geometric) equations. Rather, the Jacobian is retrieved from a LWPR model that was trained to predict task position from articulatory position. Therefore, we calculate a rough estimate of the derivative by computing the midpoint of the Jacobian at the current time step and the Jacobian at the previous time step:

$$\dot{\mathbf{J}}(\tilde{a}_t, \dot{\tilde{a}}_t) = \frac{\mathbf{J}(\tilde{a}_t) - \mathbf{J}(\tilde{a}_{t-1})}{0.005} \tag{5}$$

where the denominator, 0.005, refers to the duration of each time step in seconds (i.e., 5 ms).

The third term, $(\mathbf{I}_n - \mathbf{J}(\tilde{a}_t)^\dagger \mathbf{J}(\tilde{a}_t))\ddot{a}_d$, is a supplementary dissipative acceleration term that is implemented to remove non-negligible and undesirable velocities [48]. This term is referred as null projection term, because it works as an orthogonal projection operator to remove movements in the null space. The $\mathbf{I}_n$ is an $n \times n$ identity matrix in which $n$ equals 7 to match the task position dimension. This null projection term also includes $\ddot{a}_d$ which is the damping acceleration vector:

$$\ddot{a}_d = B_N \dot{\tilde{a}}_t \tag{6}$$

The neutral attractor damping term, $B_N$ is also used for the last term of the articulatory motor command, the neutral attractor [48]:

$$\ddot{a}_N = G_N(-B_N \dot{\tilde{a}}_t) - K_N(\tilde{a}_t - a_N) \tag{7}$$

which serves the purpose of moving any non-active articulators (i.e., not assigned to move based on the gestural score) towards a pre-defined neutral position. The gating vector, $G_N$, makes sure that only the non-active articulators are chosen in this term. The neutral damping term, $B_N$, and stiffness term, $K_N$, are predefined as global variables (*globalvariables.py*). Lastly, $a_N$ is the neutral (resting) position defined as a global variable, which was always set to be the same as initial position in the study. The resulting articulatory motor command is sent to both the plant as well as the articulatory state estimator (articulatory efference copy).

**Plant (*ArticKinematics.py, AcousticSynthesis.py, Maeda.py*).** In this module, a Gaussian plant noise is first added to the articulatory motor command. The plant noise reflects a combination of various sources of noise that exist in the neurophysiology of the speech motor system (e.g., cellular, electrical/action potentials, synaptic, motor noise, see Faisal et al. [123] for in-depth discussion). The amount of plant noise is determined by the plant noise scale specified in the configuration file (default 0.05), which is multiplied by a normalization vector to allow users to control the amount of noise added to each Maeda parameter.

The articulatory motor command, combined with noise, is then integrated using the *solve_ivp* function, an Ordinary Differential Equation (ODE) solver included in the *scipy.integrate* sub-package [124]. This function is called along with various input parameters in order to specifically mimic the ODE45 function in Matlab used in [38]. The integration requires the true articulatory state of the previous step, $\mathbf{a_{t-1}}$.

The resulting articulatory state, $\mathbf{a_t}$, is sent to *AcousticSynthesis* class which adds sensory noise. Importantly, this module also generates acoustic output (i.e., formant frequencies). The module relies on the plant model, Maeda, whose acoustic output becomes auditory signals. During the initial simulations, we realized that F1 and F2 outputs of the model had a minimum resolution of $\sim 8.5$ Hz. This resolution was not satisfactory given that some contexts (i.e., online compensation) required a magnitude of changes smaller than the Maeda model's default formant resolution. Therefore, we updated the original Maeda source code to search for the spectral peaks with a higher resolution (see S2 Appendix). The modified Maeda (High resolution Maeda or HR Maeda) is capable of producing F1 and F2 at 1 Hz resolution. The HR Maeda was wrapped in Python using an open-source software tool, Simplified Wrapper and Interface Generator (SWIG) [125]. The higher resolution auditory signals (i.e., F1, F2, and F3) are fed into the *AuditoryPerturbation* class as described below.

**Auditory perturbation (*AuditoryPerturbation.py*).** Auditory feedback can be perturbed either in Hz or cents depending on the perturbation mode set in a given configuration file. The specific time steps and trials can be set to be perturbed from the configuration file. The resulting perturbed (or unperturbed) auditory feedback is then picked up by auditory system in the *SensorySystemNoise* class.

**Sensory noise (*SensorySystemNoise.py*).** Both somatosensory and auditory signals are added with noise. Each noise is specified by noise scale factors that can be specified in the configuration file. In addition, a normalization factor giving default values for each formant is used to scale the auditory noise to a relevant magnitude. By default, the vector is set as [532 Hz, 1759 Hz, 2589 Hz], corresponding to a typical male speaker's production of F1, F2, and F3 of the vowel (/ε/) respectively. The somatosensory noise is also added in a similar manner.

**State estimators (*ArticStateEstimator.py, TaskStateEstimator.py*).** The FACTS state estimators are designed with Unscented Kalman Filter (UKF) processes [38]. The first step of

UKF is applying unscented transform of the state and computing the mean and covariance of the (transformed) sigma points. In this section, the Kalman filtering processes and variables will not be described in detail except for some instances of multiplying gains to several covariance matrices of the unscented transformed sigma points (i.e., AUKF). For more details on the general UKF processes, see [55] or [38].

After the unscented transform, the mean sigma points of the articulatory state ($\tilde{\mathbf{a}}_{t-1}$, note that we did not change the notation for unscented transformed variables for simplicity) become an input to the articulatory state prediction ($\mathcal{F}_{\mathbf{a}}$), a forward model that predicts the articulatory state:

$$\hat{\mathbf{a}}_t = \mathcal{F}_{\mathbf{a}}(\tilde{\mathbf{a}}_{t-1}, \ddot{a}_t) \tag{8}$$

The prediction also requires the efference copy of articulatory motor command ($\ddot{a}_t$). In Design C, the articulatory prediction $\hat{\mathbf{a}}_t$ is then used to predict somatosensory feedback through the somatosensory prediction LWPR model ($\mathcal{H}_{somato}$):

$$\hat{y}_t^{somato} = \mathcal{H}_{somato}(\hat{\mathbf{a}}_t) \tag{9}$$

The resulting predicted somatosensory feedback is compared with the actual somatosensory feedback to compute the somatosensory prediction error:

$$\Delta y_t^{somato} = y_t^{somato} - \hat{y}_t^{somato} \tag{10}$$

For state correction, the somatosensory prediction error is then multiplied by a Kalman gain ($\mathcal{K}\Delta y_t^{somato}$). The state correction term is added to the original state prediction to determine the final articulatory state estimate:

$$\tilde{\mathbf{a}}_t = \hat{\mathbf{a}}_t + \mathcal{K}\Delta y_t^{somato} \tag{11}$$

In Design A and B, the articulatory state estimator also makes use of the auditory prediction model ($\mathcal{H}_{\mathbf{aud}}$) to predict auditory feedback, in addition to the the somatosensory prediction ($\mathcal{H}_{\mathbf{somato}}$). In these designs, the task state estimator is implemented as the articulatory-to-task state transformation LWPR (*Task_model*). Hence, the task state estimator would "estimate" task state based on the current articulatory state estimate [38].

In Design C, the task state estimator is designed as a separate state estimator that uses auditory feedback. The first step of the task state estimator also transforms the articulatory state estimate into the task state using the same LWPR model (*Task_model*), but the articulatory state estimate of the previous time step is transformed into the task state estimate of the previous time step:

$$\tilde{\mathbf{x}}_{t-1} = Task\_model(\tilde{\mathbf{a}}_{t-1}) \tag{12}$$

The state estimate from the previous time step ($\tilde{\mathbf{x}}_{t-1}$) can then be used for the task state prediction ($\mathcal{F}_{\mathbf{x}}$):

$$\hat{\mathbf{x}}_t = \mathcal{F}_{\mathbf{x}}(\tilde{\mathbf{x}}_{t-1}, \ddot{x}_t) \tag{13}$$

where $\ddot{x}_t$ is the efference copy of the task motor command. Based on the resulting task state prediction ($\hat{x}_t$), auditory prediction is generated:

$$\hat{y}_t^{aud} = \mathcal{H}_{\mathbf{aud}}(\hat{\mathbf{x}}_t) \tag{14}$$

which can be subtracted from the actual auditory feedback to compute auditory prediction

errors (auditory prediction errors):

$$\Delta y_t^{aud} = y_t^{aud} - \hat{y}_t^{aud} \tag{15}$$

The resulting auditory prediction errors can be used to compute state correction ($\mathcal{K}_2 \Delta y_t^{aud}$), which is added to the task prediction to determine the final task state estimate ($\tilde{\mathbf{x}}_t$):

$$\tilde{\mathbf{x}}_t = \hat{\mathbf{x}}_t + \mathcal{K}_2 \Delta y_t^{aud} \tag{16}$$

where $\mathcal{K}_2$ denotes the Kalman gain for task state corrections. Note that after the last time step for each utterance, the articulatory state estimates ($\tilde{\mathbf{a}}_{t-1}$) and their corresponding final task state estimates ($\tilde{\mathbf{x}}_t$) saved in the error memory buffer are used to update the task transformation LWPR (see the LWPR training and update section for more details).

For task state corrections involving $\mathcal{K}_2$, a simple AUKF design is implemented. Specifically, the AUKF would either be activated or deactivated depending on the magnitude of $\epsilon$, which is the square of the auditory prediction errors, inversely weighted by the auditory noise covariance matrix:

$$\epsilon = \Delta y_t^{aud\top} \mathbf{P}_{\mathbf{yy}}^{-1} \Delta y_t^{aud} \tag{17}$$

where $\mathbf{P}_{\mathbf{yy}}$ refers to sensory (auditory) sigma points covariance matrix. If this term is large, this means that the Kalman filter is expecting large variance in the auditory feedback, which would make the filter less sensitive to auditory prediction errors. This variance term is used as an inverse weight (i.e., $\mathbf{P}_{\mathbf{yy}}^{-1}$) so that the resulting $\epsilon$ can account for the variance in the auditory system in addition to the size of auditory prediction errors. Whenever $\epsilon$ is larger than $\gamma$ which is set to 50 for all simulations in the study, the AUKF is activated by multiplying three UKF filter terms by fixed scales such that:

$$\mathbf{P}_{\mathbf{xx}}(t-1) = \begin{cases} \beta_{xx} \mathbf{P}_{\mathbf{xx}}^{\mathbf{def}} & \text{if } \epsilon > \gamma \\[2ex] \mathbf{P}_{\mathbf{xx}}^{\mathbf{def}} & \text{otherwise} \end{cases}$$

$$\mathbf{Q}(t-1) = \begin{cases} \beta_Q \mathbf{Q}^{\mathbf{def}} & \text{if } \epsilon > \gamma \\[2ex] \mathbf{Q}^{\mathbf{def}} & \text{otherwise} \end{cases} \tag{18}$$

$$\mathbf{P}_{\mathbf{xy}}(t) = \begin{cases} \beta_{xy} \mathbf{P}_{\mathbf{xy}}^{\mathbf{def}} & \text{if } \epsilon > \gamma \\[2ex] \mathbf{P}_{\mathbf{xy}}^{\mathbf{def}} & \text{otherwise} \end{cases}$$

The superscript ($^{\mathbf{def}}$) refers to default matrices in non-adaptive UKF. $\mathbf{P}_{\mathbf{xx}}$ denotes the prior covariance matrix of the task state estimate that is used to determine the size of the spread in the sigma points during unscented transformation. $\mathbf{Q}$ represents the process noise scale, which is added in the prior covariance matrix. Multiplying these terms by a scaling factor does not change the model behavior in the given time step, because the unscented transformation has already been computed. Instead, the adapted (i.e., multiplied) terms will increase the spread of the next time step's sigma points, which would cause the filter to be more sensitive to auditory feedback (i.e., larger Kalman gain) in that time step. The last variable listed ($\mathbf{P}_{\mathbf{xy}}$) in the Eq (18) is the cross covariance of the task state and auditory feedback. This variable, on the other

hand, is directly involved in Kalman gain calculation in the current time step:

$$\mathcal{K}(t) = \mathbf{P}_{xy}(t)\mathbf{P}_{yy}^{-1}(t) \tag{19}$$

Whenever the $\epsilon$ is not larger than $\gamma$, the AUKF mode is deactivated, and the filter would act as the regular (non-adaptive) UKF with its default variable values. In many applications of the AUKF, the filter offers more versatility by setting AUKF scaling factors to be dependent on $\epsilon$. In contrast, we use simple fixed (i.e., non-changing) gain factors ($\beta_{xx}, \beta_Q, \beta_{xy}$) to multiply those terms as this is the first attempt to implement AUKF in the state feedback control/FACTS model. Nevertheless, we found that this simple AUKF implementation was able to generate realistic adaptation simulations.

## LWPR training and update

The LWPR source code was wrapped in Python 3.6.10 using SWIG [125]. In order to train the LWPR models, a large data set of Maeda model inputs and outputs were generated via a random walk algorithm (see a similar data set in Gaines et al. [80]). For each articulatory configuration, we extracted the corresponding task parameters from the Maeda vocal tract shape coordinates' horizontal axis (see Fig 8). Thus, the random walk approach produced a near-complete data set of Maeda input parameters (see S2 Appendix for more details) and their corresponding task state as well as formant values. These allowed us to train the articulatory state prediction ($\mathcal{F}_a$), task state prediction ($\mathcal{F}_x$ in Design C), somatosensory prediction ($\mathcal{H}_{somato}$), and auditory prediction ($\mathcal{H}_{aud}$) LWPR models. For each LWPR model, we trained several prototype models and chose a model that produced the most accurate results. The final LWPR models were saved in binary format files, which could be called naïve (i.e., not adapted) models. Once these models are loaded in FACTS simulations, they can be continuously updated in response to auditory perturbation using the built-in LWPR update method (*LWPR.update*). Nevertheless, the updated LWPR models do not overwrite the binary format files so that each simulation run can call and use the naïve LWPR models.

During adaptation simulations, each LWPR module is updated by using its built-in function. All time steps saved in the error memory are used to update a given LWPR module (e.g., articulatory-to-task transformation in Design C). There are LWPR parameters that can be tuned to affect learning to a degree, but they also affect the stability of LWPR models much like the stability versus plasticity dilemma (e.g., [126]). Thus, these parameters cannot be changed drastically, so all simulations were done with fixed LWPR parameters that would make the LWPR model stable. More details on these parameters can be found in S3 Appendix.

## Simulations

All simulations of adaptation generated in the current study involved repeated productions of the vowel, /ɛ/, which were produced for 40 time steps (200 ms). /ɛ/ was chosen because the vowel is one of the most extensively studied vowels in F1 adaptation studies. It should be noted that adaptive mechanisms in the FACTS model have no linguistic constraints, but a thorough exploration of various vowels in adaptation remains to be investigated both experimentally and computationally (see Discussion).

The vowel was produced using a gestural score with a target of 22.1 mm constriction degree for the tongue body at the hard palate, driving the tongue to move towards this state from from its initial position. The task target (22.1 mm) defined in the gestural score file was chosen based on the fact that the corresponding first formant frequency (530 Hz) was close to the male speakers' F1 production for the vowel [127]. In these simulations, we wanted the model to quickly achieve the task target from the initial position so that steady-state vowel portion

composed most of the utterance. For that reason, the initial position (22.19 mm) was configured to be very close to the target vowel.

It is understood that auditory feedback-based corrections are minimal, if any, in short vowel productions (i.e., < 150 ms). Hence, the adaptation simulations were designed to ignore sensory feedback-based corrections (i.e., no online compensation). Instead, auditory feedback or its related information (i.e., the final state estimate updated by state correction signals in Design A and C) were stored in an "error memory" if the auditory prediction errors were larger than a given threshold. This memory was accessed at the end of each trial to update the model in the subsequent production. In order to rule out the possibility that our findings are limited to the situations without online compensation, however, we also tested all simulations with online compensation and confirmed that these results were comparable with the simulations presented in the study.

The first 10 time steps were designed to simulate non-vocalized movement (e.g., initial jaw opening from the resting position before the voicing of the vowel). Hence, for each production, we considered only the latter 30 time steps as (simulated) acoustic data. For plotting adaptation, an average of each production was computed from the mid-utterance time steps (i.e., middle 10 time steps of the 30 time steps) of "acoustic" data.

In all simulations, the auditory perturbation was configured to match the experimental setup. For example, 400 cents up-shift in F1 was applied in simulations compared with Kim & Max [19]. Likewise, simulations for Mitsuya et al. [61] had the perturbation increased or decreased by 4 Hz each trial as specified in the study. For Mollaei et al. [62], the simulation applied 454 cents up-shift to match with the 30% in Hz up-shift perturbation in the study. Some simulation results were normalized by either subtracting the baseline F1 (530 Hz) or dividing by the baseline F1 and F2 values (530 Hz and 1775 Hz respectively) as in Fig 4C. Because fitting FACTS simulations to empirical data was not the main scope of the study, only a few model parameters were slightly changed by visual inspections and trial-and-error attempts. Here, we presented the simulations run with parameters that best mimicked the empirical data, which are listed in Table 2.

For the online compensation simulation (Fig 6), the same vowel was used, so most of the gestural score parameters were kept the same (e.g., task target). However, the length of the production was set to be longer (80 time steps) in order to simulate the mid-utterance perturbation. Another difference from the adaptation simulations was that the AUKF was not used with reasons described in the Results section. All parameters of FACTS remained the same as in the simulation adaptation except the auditory noise level, which was doubled (Table 2). All FACTS simulations presented in the current study can be found on an OSF data repository (DOI: 10.17605/OSF.IO/W37HV).

## Experimental data used in comparisons with the simulations

All experimental data were retrieved from studies involving the same vowel (/ɛ/) production (or monosyllabic utterances with the same vowel) as our simulations. The data set included in Figs 3, 4, 7, and S3 Appendix was the control group's (i.e., people who do not stutter) average data in Kim & Max [19]. The data set was retrieved from the text files available on the study's Open Science Framework (OSF) data repository. For our model simulations, the empirical data was converted from Cents to Hertz using the baseline value of 530 Hz by:

$$F_{Hz} = 530 \times 2^{\frac{F_{cents}}{1200}} \tag{20}$$

In addition, the experimental data set used in Fig 4 was from the English native speakers reported in Mitsuya et al. [61]. The group average data was extracted from the study by

**Table 2. The tuning of various parameters for simulations shown in the figures.** The first three parameters listed ($\beta_{xx}$, $\beta_Q$, $\beta_{xy}$) were configured for AUKF, each functioning as multiplicative gains for prior covariance of task state sigma points ($\mathbf{P_{xx}}$), process noise ($\mathbf{Q}$), and cross covariance of the task state and the auditory feedback sigma points ($\mathbf{P_{xy}}$) in the cases that AUKF is activated. Note that the gains were 1 for the simulations presented in Figs 3 and 6 because AUKF was not enabled. The other two parameters changed in some simulations were auditory noise size ($\eta_{aud}$) as well as auditory prediction error threshold ($\gamma$).

|  | Prior ($\beta_{xx}$) | Process ($\beta_Q$) | Cross ($\beta_{xy}$) | Aud. N. ($\eta_{aud}$) | Thresh. ($\gamma$) Hz | Targ. N. ($\eta_{targ}$) mm |
|---|---|---|---|---|---|---|
| Fig 3 Design C | 1 | 1 | 1 | 0.005 | 5 | 0 |
| Fig 4B (Top) | 10 | 10 | 6 | 0.005 | 5 | 0 |
| B (Bottom) | 4 | 4 | 4 | 0.005 | 35 | 0 |
| C | 2 | 2 | 2 | 0.005 | 20 | 0 |
| Fig 6 | 1 | 1 | 1 | 0.01 | N/A | N/A |
| Fig 7A | 6 | 6 | 6 | 0.005 | 5 | 0 |
|  | 3 | 3 | 3 | 0.005 | 5 | 0 |
|  | 1.5 | 1.5 | 1.5 | 0.005 | 5 | 0 |
| Fig 7B | 10 | 10 | 6 | 0.005 | 5 | 0 |
|  | 10 | 10 | 6 | 0.005 | 30 | 0 |
|  | 10 | 10 | 6 | 0.005 | 50 | 0 |
| Fig 7C | 10 | 10 | 6 | 0.0025 | 5 | 0 |
|  | 10 | 10 | 6 | 0.005 | 5 | 0 |
|  | 10 | 10 | 6 | 0.01 | 5 | 0 |
| Fig 7D | 10 | 10 | 6 | 0.005 | 5 | 0 |
|  | 10 | 10 | 6 | 0.005 | 5 | 0.2 |
|  | 10 | 10 | 6 | 0.005 | 5 | 0.4 |

WebPlotDigitizer, a free web-based data extraction tool [128]. Lastly, a data set came from the control group's (i.e., people without Parkinson's disease) average data in Mollaei et al. [62], which was also extracted using WebPlotDigitizer.

## Supporting information

**S1 Appendix. Additional analyses on Design A.**
(DOCX)

**S2 Appendix. High Resolution Maeda.**
(DOCX)

**S3 Appendix. LWPR forgetting factors.**
(DOCX)

## Acknowledgments

We thank Dr. Satrajit Ghosh for his guidance on the Maeda model.

## Author Contributions

**Conceptualization:** Kwang S. Kim, Jessica L. Gaines, Benjamin Parrell, Vikram Ramanarayanan, Srikantan S. Nagarajan, John F. Houde.

**Data curation:** Srikantan S. Nagarajan.

**Formal analysis:** Kwang S. Kim, Jessica L. Gaines, Vikram Ramanarayanan.

**Funding acquisition:** Kwang S. Kim, Srikantan S. Nagarajan, John F. Houde.

**Investigation:** Kwang S. Kim, Jessica L. Gaines, Benjamin Parrell, Vikram Ramanarayanan, Srikantan S. Nagarajan, John F. Houde.

**Methodology:** Kwang S. Kim, Jessica L. Gaines, Benjamin Parrell, Vikram Ramanarayanan, Srikantan S. Nagarajan, John F. Houde.

**Project administration:** Benjamin Parrell, Vikram Ramanarayanan, Srikantan S. Nagarajan, John F. Houde.

**Resources:** Benjamin Parrell, Vikram Ramanarayanan, Srikantan S. Nagarajan, John F. Houde.

**Software:** Kwang S. Kim, Jessica L. Gaines, Benjamin Parrell, Vikram Ramanarayanan, Srikantan S. Nagarajan, John F. Houde.

**Supervision:** Benjamin Parrell, Vikram Ramanarayanan, Srikantan S. Nagarajan, John F. Houde.

**Validation:** Kwang S. Kim, Jessica L. Gaines, Benjamin Parrell, Vikram Ramanarayanan, Srikantan S. Nagarajan, John F. Houde.

**Visualization:** Kwang S. Kim, Benjamin Parrell, Vikram Ramanarayanan, Srikantan S. Nagarajan, John F. Houde.

**Writing – original draft:** Kwang S. Kim.

**Writing – review & editing:** Kwang S. Kim, Jessica L. Gaines, Benjamin Parrell, Vikram Ramanarayanan, Srikantan S. Nagarajan, John F. Houde.

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
