## [Decision Letter · Decision Letter 0]

29 Nov 2022

Dear Dr. Kim,

Thank you very much for submitting your manuscript "Mechanisms of sensorimotor adaptation in a hierarchical state feedback control model of speech" for consideration at PLOS Computational Biology.

As with all papers reviewed by the journal, your manuscript was reviewed by members of the editorial board and by several independent reviewers. In light of the reviews (below this email), we would like to invite the resubmission of a significantly-revised version that takes into account the reviewers' comments.

Dear Kwang and colleagues,

You will find a thorough review by two experts in the field of your manuscript. Note that the GitHub link you provided was not functional. We look forward to your detailed responses.

Best wishes,

Frederic Theunissen

We cannot make any decision about publication until we have seen the revised manuscript and your response to the reviewers' comments. Your revised manuscript is also likely to be sent to reviewers for further evaluation.

Sincerely,

Frédéric E. Theunissen

Academic Editor

PLOS Computational Biology

Thomas Serre

Section Editor

PLOS Computational Biology

Dear Kwang and colleagues,

You will find a thorough review by two experts in the field of your manuscript. Note that the GitHub link you provided was not functional. We look forward to your detailed responses.

Best wishes,

Frederic Theunissen

Reviewer's Responses to Questions

**Comments to the Authors:**

Reviewer #1: See attached file

Reviewer #2: This study refines the authors’ previous computational model of speech (FACTS) in order to account for a range of phenomena in sensorimotor adaptation of speech. A number of versions of the model are proposed, with ones incorporating a task update module being able to yield adaptation patterns similar to the ones seen in literature. This is an interesting model potentially very useful to the field. Limitations to the model are fairly outlined and discussed. Resolving some of them would make for a more comprehensive and helpful model (e.g. modeling how and why adaptation plateaus); however, the model formulations proposed here already provide interesting insights. I found parts of the modeling/results a bit hard to read – minimizing jargon and adding clarifications could help the reader (especially people outside the field, or less familiar with this kind of computational modeling) follow this work more easily. I have a few questions and requests for clarification below:

(1) The introduction ends with stating that the paper will “Examine the idea that adaptation is driven by sensory prediction errors” – presenting it as the main question. However, that’s not what’s yet answered here - the model does show how sensory prediction errors could drive speech adaptation, but it does not consider alternatives to that. I think this work would benefit from a discussion of how other kinds of errors (such as task errors) could play a role. For example, in the reaching adaptation literature, there is evidence that task errors are involved in adaptation – for example, by interacting/contributing to implicit visuomotor adaptation along with sensory prediction errors (Leow et al., EJN 2018); more recent work suggested that task errors could drive adaptation on their own (Lew et al., J Neurosci 2000; Rajan et al, “motor performance prediction error” MLMC meeting 2000), without the need for sensory prediction errors. Would be useful to discuss how the model could distinguish which kind of errors drives speech adaptation, either in its current formulation or in the future.

(2) It doesn’t seem surprising that the original formulation – and its two variants, A and B – would not yield adaptation by sensory prediction error: if the forward models (updated through acoustic error as in B, or not as in A) do not include a way for the updated prediction to translate to changed motor output, how could one get any adaptive changes in output?

Related to the above: Figure 3, design A: what is the source/mechanism behind the (small) shift in model output here?

(3) The point that simulation results are in line with direct policy adaptation (lines 492-495) is interesting – but isn’t that a natural consequence of the model design? The model takes SPE and directly adapts the policy at the task level. By removing updates to the forward model, the SPE never gets reduced, resulting in policy updates that would be, in theory, unlimited (i.e. without any forgetting factors or other way to have a saturation limit, adaptation would keep going beyond the target frequency – akin to an error-clamp in reaching adaptation (Morehead et al., 2017). To distinguish between forward- vs. policy-based adaptation, it would be fair to simulate a true forward-based model as well – where the prediction is updated, and then “inverted” somehow to yield the motor command. I thought that was the articulatory feedback control law would do: I’m a bit puzzled how changing the articulatory feedback control law doesn’t yield adaptation (Figure 5). Isn’t the control law the inverse of the articulatory-to-task transformation, thus assuming that the articulatory-to-task prediction is properly adapted, inverting it would yield adaptive changes at the task level?

(4) Was there any reason behind using only the sound /ε/ for simulations? How robust would the model be in predicting adaptive changes when other sounds are being perturbed?

Other points

Line 329: missing citation of commonly reported online compensation response

Lines 599-601: That’s a valid argument, but an explicit report does not necessarily mean that what is being reported was the result of an explicit process.

Lines 609-610: in FACTS design… in FACTS (repeated)

Lines 707-711: Does online compensation (in absolute terms) decrease, or does it decrease relative to the perturbation size?

Lines 818-820: repeating the previous sentence

Repository link doesn’t seem to work

A list of abbreviations (or reducing their number) may be helpful

**Have the authors made all data and (if applicable) computational code underlying the findings in their manuscript fully available?**

Reviewer #1: Yes

Reviewer #2: **No: **There's a link to a github repository, there's nothing there yet

PLOS authors have the option to publish the peer review history of their article (what does this mean?). If published, this will include your full peer review and any attached files.

Reviewer #1: **Yes: **Pascal Perrier, Gipsa-lab, Université Grenoble Alpes, Grenoble INP, CNRS, France

Pierre Baraduc, Gipsa-lab, Université Grenoble Alpes, Grenoble INP, CNRS, France

Reviewer #2: No
---

## [Decision Letter · Decision Letter 1]

1 May 2023

Dear Kim,

Thank you very much for submitting your manuscript "Mechanisms of sensorimotor adaptation in a hierarchical state feedback control model of speech" for consideration at PLOS Computational Biology. As with all papers reviewed by the journal, your manuscript was reviewed by members of the editorial board and by several independent reviewers. The reviewers appreciated the attention to an important topic. Based on the reviews, we are likely to accept this manuscript for publication, providing that you modify the manuscript according to the review recommendations.

Dear Kwang and colleagues,

As you will read the two reviewers appreciated the effort you put in for the revision and stress the merit of your work. Reviewer 1 still has substantial reservations on the significance of "Design B" and potential over interpretation of your null result. The reviewer suggests eliminating that part of the simulation from paper. On the other hand, there are now additional details in the supplementary material that could be included in the main paper. This somewhat straightforward organization of your results could increase the impact of your work. I will read your reply and opinion carefully.

Looking forward to it.

Best,

Frederic Theunissen

Sincerely,

Frédéric E. Theunissen

Academic Editor

PLOS Computational Biology

Thomas Serre

Section Editor

PLOS Computational Biology

Dear Kwang and colleagues,

As you will read the two reviewers appreciated the effort you put in for the revision and stress the merit of your work. Reviewer 1 still has substantial reservations on the significance of "Design B" and potential over interpretation of your null result. The reviewer suggests eliminating that part of the simulation from paper. On the other hand, there are now additional details in the supplementary material that could be included in the main paper. This somewhat straightforward organization of your results could increase the impact of your work. I will read your reply and opinion carefully.

Looking forward to it.

Best,

Frederic Theunissen

Reviewer's Responses to Questions

**Comments to the Authors:**

Reviewer #1: I would like to thank the authors for their very interesting responses to the points raised in the review of the preceding manuscript. The changes that they have provided to the text improve the quality of the paper and give place to a more objective and nuanced interpretation of their results. The responses have in particular shed light on the behavior of Design A, in which the absence of adaptation was quite surprising. These explanations should be better integrated in the manuscript, and the discussion should consider them more explicitly. In addition, these insights raise new questions about Design A and Design C that should also be clarified. This is why major modifications are still required, before the manuscript can be accepted for publication.

First of all, reading the comments of the second reviewer, and the answers of the authors to the both reviews, I think that we all agree that Design B is scientifically not relevant. Anyone minimally aware of how SFC works can predict the results of Design B in the absence of a task specification in the auditory domain. This result does not demonstrate that it is wrong to assume that “prediction-based forward model updates alone may result in learning (see discussions in Hadjiosif et al., 2021)” (Citation from the authors’ response). Indeed if the task is defined in the auditory domain, as shown for example in Patri et al., (2018, Plos Comp. Biol.) prediction-based forward model updates does result in learning. Hence, I think that Design B should be removed from the manuscript. It adds unnecessarily to its complexity, and removing it will avoid statements that are too rapid and then inaccurate as in line 185 “These results clearly indicate that updates in auditory prediction did not generate adaptation.”

More interesting is the analysis of the results obtained with Design A.

A first very important result that should absolutely be presented in the text, and not in Supplementary Material (S1 in this case), is the key-role of the somatosensory input in the observed absence of adaptation. This result shows that the key problem in Design A is not the fact that adaptation primarily influences the predicted articulatory state, but that (1) the predicted articulatory state is satisfactory for one of the sensory comparison (the somatosensory one) and (2) the weight of this input is too large to let the auditory prediction error induce significant changes in the articulatory state. Clearly then, adaptation would happen in Design A, if the weight of the auditory modality would be larger than it is in the current implementation. This statement is all the more important since Design C not only integrates a prediction of the Task State, but also distributes the role of the somatosensory and auditory comparison over two different correction mechanisms: the somatosensory comparison, which results in no error, applies to the predicted articulatory state only, whereas the auditory comparison, which results in an error, applies to the predicted task state. It would be interesting to test what would happen in Design C, if the task state would be corrected under the influence of both sensory modalities….

A second important result is the fact that when auditory feedback alone in taken in consideration the adaptation observed in Design A is too slow. This slow adaptation is also observed in the predicted articulatory state in the middle left panel of Fig. 3, which explains why the predicted auditory output (top panel, pink curve) does not change much, since it is measured on the very beginning of the produced vowel. This slow adaptation has a striking similarity with the slow adaptation of Design C (Fig3, top right panel, blue curve), which the authors note is at odds with the experimental data (black curve), and which they corrected by replacing the usual Kalman filter design by the adaptive uscented Kalman filter (AUKF) design (Fig. 4., panel B). What would happen in Design A if AUKF would be used? This is an interesting question that should be tested…

In sum the information that the authors provided in their response about the behavior of Design A is crucial, and cannot be put as an additional information. It has to be in the main text. It clearly suggests that adaptation can occur under certain conditions with Design A. For this adaptation to occur, it requires a certain weighting between the somatosensory and the auditory feedback, and this is an issue that has been already addressed in many papers in the context of the DIVA model, the Bayesian GEPPETO model or of experimental studies such the one of Feng et al. (2011) or Lametti et al. (2012). Now if the speed of the adaptation is not consistent with experimental observations, independently of the design of the Kalman filtering, then this is an important point that speaks in favor of Design C. However, if it is the case, given the fact that, as nicely acknowledged in the revised manuscript in lines 456 to 462, DIVA and GEPPETO demonstrated adaptation in responses to auditory feedback perturbation, when internal models were updated and the task was defined in the auditory domain, it would be probably more correct to change the main conclusion. Indeed the work would then rather show that adaptation is more efficient when updates affect internal models that the task state, rather than any other signal that is not directly related to the task. In this case, the abstract, the discussion and the conclusion should be modified accordingly.

The question raised by Reviewer 2 about the absence of adaptation when the articulatory feedback control law changes in Fig. 5 and the explanation provided by the authors in Supplementary Material S2 should also be clearly integrated in the text. Indeed it shows that the absence of adaptation is not associated with the principle of a change of the articulatory feedback control law per se but with its specific implementation in the context of the Maeda’s model and the computation of the Jacobian matrix used by the authors….. In my opinion the sentence in line 317-320 should then be modified to be clear about this point, and exclude any kind of generalization to other models, even to SFC models that would use another articulatory model or compute the Jacobian Matrix differently.

Reviewer #2: Thank you for addressing my comments. There definitely interesting issues to be explored further: examples include the issue of saturation of adaptation, or whether using a different/more robust way to invert the articulatory-to-task transformation (discussion in Supplementary materials 2) would change the outcome in Figure 5 Right. However, I find my comments adequately addressed, and that the paper does a good job presenting this interesting model and fairly discussing its limitations.

**Have the authors made all data and (if applicable) computational code underlying the findings in their manuscript fully available?**

Reviewer #1: Yes

Reviewer #2: Yes

PLOS authors have the option to publish the peer review history of their article (what does this mean?). If published, this will include your full peer review and any attached files.

Reviewer #1: No

Reviewer #2: No

Figure Files:

Data Requirements:

Reproducibility:

References:

---

## [Editor Report · Decision Letter 2]

6 Jun 2023

Dear Kim,

We are pleased to inform you that your manuscript 'Mechanisms of sensorimotor adaptation in a hierarchical state feedback control model of speech' has been provisionally accepted for publication in PLOS Computational Biology.

Best regards,

Frédéric E. Theunissen

Academic Editor

PLOS Computational Biology

Thomas Serre

Section Editor

PLOS Computational Biology

Dear Kwang Kim,

Thank you for addressing the final moments and congratulations on a nice contribution. I agree with you that keeping design B in the manuscript can be useful for readers who are not familiar with state control feedback in speech production.

Best,

Frederic Theunissen

---

## [Editor Report · Acceptance letter]

20 Jul 2023

PCOMPBIOL-D-22-01400R2 

Mechanisms of sensorimotor adaptation in a hierarchical state feedback control model of speech

Dear Dr Kim,

I am pleased to inform you that your manuscript has been formally accepted for publication in PLOS Computational Biology. Your manuscript is now with our production department and you will be notified of the publication date in due course.

With kind regards,

Zsofia Freund
